# A unified rodent atlas reveals the cellular complexity and evolutionary divergence of the dorsal vagal complex

Cecilia Hes[1,2], Abigail J Tomlinson[3], Lieke Michielsen[4], Hunter J Murdoch[1,5], Fatemeh Soltani[1,2], Maia V Kokoeva[1,5,6], Paul V Sabatini[1,2,5,6]*

[1]Research Institute of the McGill University Health Centre, McGill University Health Centre, Montreal, Canada; [2]Division of Clinical and Translational Research, Department of Medicine, McGill University, Montreal, Canada; [3]Department of Internal Medicine, University of Michigan, Ann Arbor, United States; [4]Department of Human Genetics, Leiden University Medical Center, Leiden, Netherlands; [5]Integrated Program in Neuroscience, Department of Medicine, McGill University, Montreal, Canada; [6]Division of Endocrinology, Department of Medicine, McGill University, Montreal, Canada

*For correspondence:
paul.sabatini@mcgill.ca

Competing interest: The authors declare that no competing interests exist.

## eLife Assessment

This manuscript applies state-of-the-art techniques to define the cellular composition of the dorsal vagal complex in two rodent species (mice and rats). The result is a **fundamental** resource that substantially advances our understanding of the dorsal vagal complex's role in the regulation of feeding and metabolism while also highlighting key differences between species. The analyses of single-cell profiling experiments in the manuscript provide **compelling** insight into the cellular architecture of the dorsal vagal complex, with potential implications for obesity therapeutics. [Editors' note: The FASTQ files of the rat snRNA-Seq data for this manuscript are not available as the authors could not locate the files after moving institutions. However, the count matrices are available to download via the Broad Single cell portal.]

**Abstract** The dorsal vagal complex (DVC) is a region in the brainstem comprised of an intricate network of specialized cells responsible for sensing and propagating many appetite-related cues. Understanding the dynamics controlling appetite requires deeply exploring the cell types and transitory states harbored in this brain site. We generated a multi-species DVC cell atlas using single-nuclei RNA-sequencing, by curating and harmonizing mouse and rat data, which includes >180,000 cells and 123 cell identities at 5 granularities of cellular resolution. We report unique DVC features such as *Kcnj3* expression in Ca⁺-permeable astrocytes as well as new cell populations like neurons co-expressing *Th* and *Cck*, and a leptin receptor-expressing neuron population in the rat area postrema which is marked by expression of the progenitor marker, *Pdgfra*. In summary, our findings demonstrate a high degree of complexity within the DVC and provide a valuable tool for the study of this metabolic center.

## Introduction

Regulating appetite in response to changing environmental conditions to maintain energy balance requires the intricate interplay of multiple organs. Across the central nervous system, multiple discrete

brain regions and cell types control appetite (*Gautron et al., 2015*). Within the brainstem, the dorsal vagal complex (DVC), comprising the area postrema (AP), nucleus of the solitary tract (NTS), and dorsal motor nucleus of the vagus, acts as a key nexus point for metabolic signals, evidenced by the dense expression of many relevant hormone receptors (*Ludwig et al., 2021*). Separately, the DVC also serves as the primary site of gut-based signals via vagal afferents (*Bai et al., 2019*; *Han et al., 2018*; *Williams et al., 2016*). Functionally, the DVC was long considered a site promoting meal termination (*Schwartz, 2006*), but growing evidence suggests it also controls long-term energy balance (*Cheng et al., 2021*).

Much of the interest in DVC neurons' role in appetite and energy balance stems from their role as therapeutic targets for obesity and anorexia in cancer cachexia (*Yanovski and Yanovski, 2014*; *Borner et al., 2018*). For many years, most pharmacotherapies failed to produce efficacious and long-lasting weight loss in obesity and increase appetite in cancer cachexia (*Advani et al., 2018*; *Huang et al., 2024*; *Cheng et al., 2022*). More recently, however, the glucagon-like peptide 1 receptor (GLP1R) agonists are eclipsing the weight loss achieved by previous generations of therapeutics (*Véniant et al., 2024*). While expressed across the central nervous system, GLP1R expression is particularly enriched in DVC neurons (*Adams et al., 2018*), and it is the DVC that mediates many of the effects of GLP1R agonists (*Huang et al., 2024*). Separately, an emerging strategy for treating cancer cachexia is inhibiting growth differentiation factor 15 (GDF15) signaling in the DVC by blocking the action of GDNF Family Receptor Alpha Like (GFRAL) (*Medic et al., 2022*), the receptor for GDF15 (*Yang et al., 2017*). Given the clinical significance of the DVC for this spectrum of conditions, understanding the molecular heterogeneity and function of this region is essential.

Traditionally, identification and ensuing study of DVC cell types was limited to known neurotransmitter and receptor-expressing cell types. For instance, the cholecystokinin and monoamine-expressing cells, demarcated by tyrosine hydroxylase (*Th*) expression, were previously shown to be non-overlapping cell populations with unique roles in appetite suppression (*Roman et al., 2016*). More recently, the advent of single-cell mRNA sequencing approaches made unbiased molecular censuses of heterogeneous populations possible. Such studies focusing on the mouse have revealed the complexity of the mouse DVC, comprised of dozens of transcriptionally distinct cell types (*Ludwig et al., 2021*; *Dowsett et al., 2021*; *Zhang et al., 2021*). Moving forward, unifying information from multiple single-nuclei RNA-sequencing (snRNA-seq) datasets of the DVC into a reference atlas is needed to more fully understand the cells within the DVC and generate a common and scalable classification for cell identities. Such a reference atlas is also needed to contrast the cell types found within the mouse to other research models, including the rat, which are prominent models for DVC biology (*Ludwig et al., 2021*; *Fortin et al., 2020*).

To address this, we generated a comprehensive snRNA-seq-based atlas of the mouse and rat DVC. This atlas features mouse and rat labeled cells with increasing granularity and an accompanying computational toolbox including the mouse and rodent atlas label transfer environment using tree-Arches. The mouse DVC atlas reveals a higher degree of cellular complexity than previously appreciated. The cellular census of the rat DVC exhibits many similarities to the mouse but also rat-specific cells including a novel population of leptin receptor and *Pdgfra*-expressing neurons localized to the AP. Using this unified atlas, we also uncover the cell types that transcriptionally respond to meal consumption which may hold the potential of suppressing appetite without gastrointestinal side effects.

## Results
### The murine DVC cell classes

To generate a de novo snRNA-seq atlas of the mouse DVC, we isolated the DVC from 30 adult C57BL/6J mice and subjected them to 10X Genomics-based snRNA barcoding and sequencing (*Figure 1A*). From the CellRanger pre-processing pipeline (*Zheng et al., 2017*), we obtained 110,167 cells of which 99,740 were retained after processing. Cell identities were initially mapped to 5 databases from different brain regions and then curated manually using 473 cell identity markers (*Supplementary files 1 and 2*). We distinguished clusters of neurons and glial, vascular, and connective tissue cells, a first layer of cell identity (*Figure 1B*; *Supplementary file 3*). We further describe a second layer of cellular resolution, which are subgroups of the first layer that contain the cell classes (*Figure 1C*; *Supplementary file 4*).

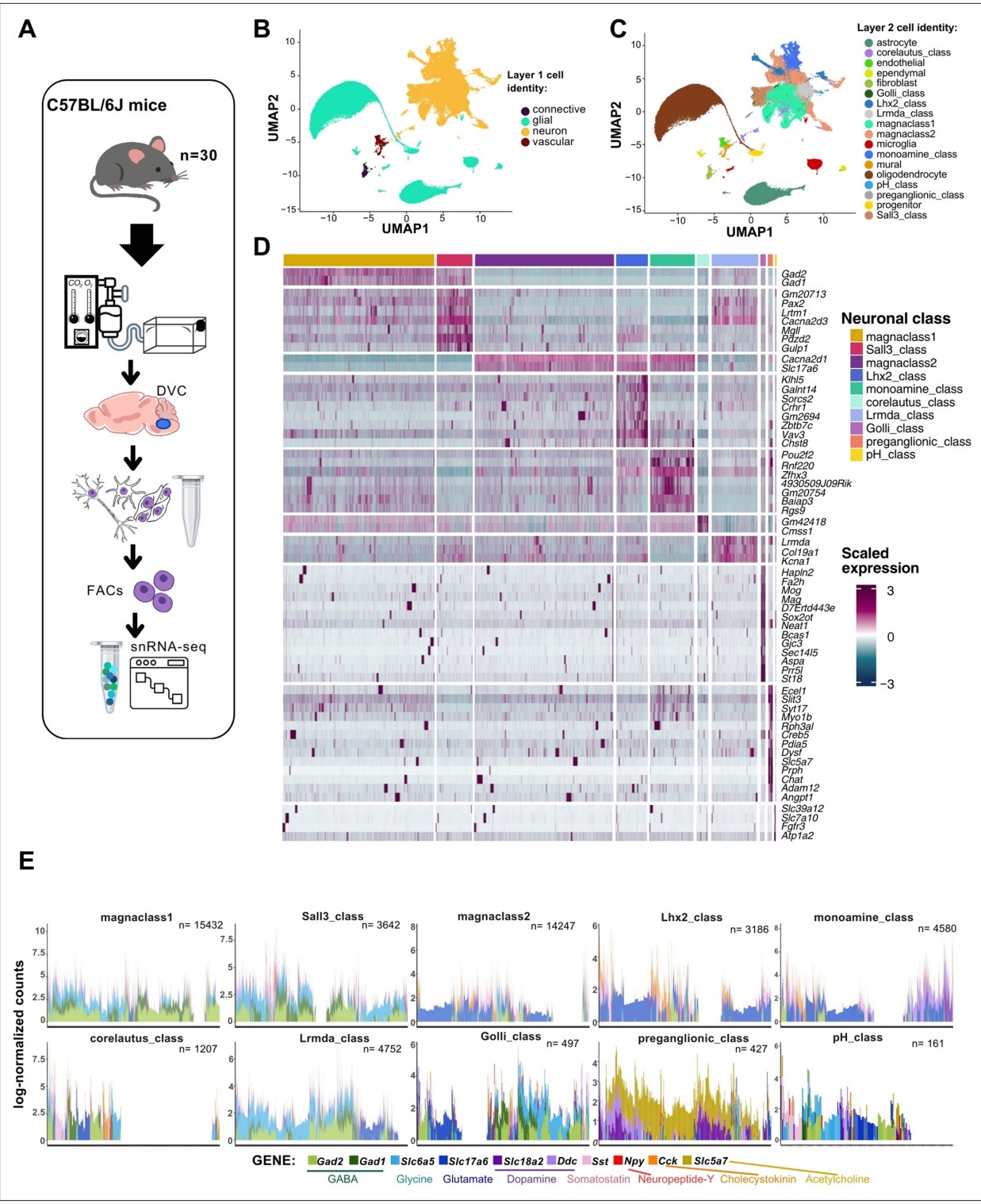

**Figure 1.** Two layers of cell identity for the murine DVC cells from snRNA-seq data. (**A**) Adult C57BL/6J mice were subjected to 10X Genomics-based snRNA barcoding and sequencing after dissection of their DVC (*n* = 30). (**B**) Labeled UMAP plot of the mouse DVC (cells *n* = 99,740) at the lowest resolution (layer-1) including 4 cell identities, and (**C**) the cell class (layer-2) including 18. (**D**) Heatmap of the main marker genes (average log₂FC expression >80 percentiles of upregulated genes) for each neuronal class of the mouse DVC (MAST algorithm; neurons *n* = 48,131; adj. p < 0.05). Each column is one cell and each row is one marker gene. (**E**) Stacked bar plot of log-normalized counts of genes for 10 transcripts (i.e. *Gad2, Gad1, Slc6a5, Slc17a6, Slc18a2, Addc, Sst, Npy, Cck,* and *Slc5a7*) related to transmission/release of 8 neurotransmitters and neuropeptides, by neuronal cell class. Each

*Figure 1 continued on next page*

*Figure 1 continued*

bar represents one neuron (*n* = 48,131). snRNA-seq = single-nuclei RNA-sequencing; DVC = dorsal vagal complex; FACS = fluorescence-activated cell sorting; UMAP = uniform manifold approximation and projection; FC = fold-change; adj. = adjusted; GABA = gamma-aminobutyric acid.

The online version of this article includes the following figure supplement(s) for figure 1:

**Figure supplement 1.** Mapping of *Gad2* and VGLUT2 in neurons.

**Figure supplement 2.** Anatomical mapping of the dorsal vagal complex (DVC) mouse layer-3 identities.

**Figure supplement 3.** Mapping of DVC populations in a murine hypothalamic dataset.

**Figure supplement 4.** Dorsal vagal complex (DVC)-centric dissection protocol.

Among the 10 neuronal cell classes identified at layer-2 resolution, we found two major groups which we called 'magnaclass 1' and 'magnaclass 2' (*Figure 1—figure supplement 1*). Magnaclass 1 neurons predominately express genes required for inhibitory neurotransmission (e.g. *Gad1* and *Gad2*) and share transcriptomic markers with the Sall3-class of neurons, whereas magnaclass 2 neurons express mostly excitatory transcripts (e.g. the glutamate transporter VGLUT2) also found in the Lhx2-class and monoamine class (*Figure 1D, E*; *Figure 1—figure supplement 1*). However, as described in other brain areas (*Vaaga et al., 2014*; *Tritsch et al., 2016*), evidence of co-transmission/co-release of multiple primary neurotransmitters was found in at least 55% of neurons regardless of their cell class (*Figure 1D, E*; *Supplementary file 5*). We also identified eight transcriptionally distinct neural classes in addition to the magnaclasses. Among these eight classes, we found two neural classes expressing traditionally non-neuronal genes: the Golli and pH-related classes. The Golli-class neurons are a relatively small cluster, comprising ~1% of all neurons. Interestingly, Golli-class neurons express multiple myelin-associated genes such as *Mag* and *Mog* (*Landry et al., 1996*; *Lutz et al., 2014*; *Cheli et al., 2016*) normally expressed by oligodendrocytes (*Figure 1D*; *Supplementary file 6*). Additionally, we identified a class of neurons with genes normally upregulated in astrocytes (e.g. *Slc6a11*) and related to clearance and uptake of glutamate, gamma-aminobutyric acid (GABA), $Ca^+$, and bicarbonate (*Melone et al., 2014*; *Theparambil et al., 2020*), which we named 'pH-related class' (*Supplementary file 6*). Similar to Golli-class neurons, pH-related neurons were a relatively rare cluster (<1% of all neurons). While the corelautus-class neurons express very few genes at higher levels, among them are *Cdk8*, *Lars2*, *Gabra6*, and *Fat2*. Expression of *Cdk8* is largely absent from the DVC but is enriched within the DVC-adjacent hypoglossal nucleus (*Figure 1—figure supplement 2*), suggesting these cells are hypoglossal neurons. We also identified the Lhx2-class, which was enriched for the Parvalbumin (*Pvalb*) gene whose expression is largely excluded from the DVC. As *Pvalb* is highly expressed in the neighboring cuneate nucleus (*Figure 1—figure supplement 2*), we posit this class may belong to this anatomically adjacent area which was captured in our dissection. When interrogating snRNA-seq data from the hypothalamus (*Steuernagel et al., 2022*; *Figure 1—figure supplement 3*), another brain site important for energy balance, we could not find clusters sharing all markers with our neuronal classes, indicating DVC-specific neuronal programs (*Figure 1—figure supplement 3*).

## The murine DVC cells at their highest resolution

To better define the heterogeneity of the mouse DVC, we further assigned a clustering-based third layer of cellular resolution resulting in fifty cell identities, of which 15 are non-neuronal (*Figure 2A*). At this resolution, we differentiated between two major groups of astrocytes. Differential expression in synapse-related factors revealed a group of astrocytes with high expression of the inwardly rectifying GIRK1 (*Kcnj3*) channel (*Kubo et al., 1993*) and lack of expression of the AMPAr subunit GluA2 (*Gria2*), additionally rendering them $K^+/Na^+/Ca^+$-permeable (*Burnashev et al., 1992*; *Geiger et al., 1995*; *Figure 2B*). We speculate these astrocytes are specific to the DVC as we could not find *Kcnj3*+; *Gria2*-astrocytes in the hypothalamus, forebrain, cortex, or hippocampus (*Steuernagel et al., 2022*; *Batiuk et al., 2020*; *Jessa et al., 2019*; *Figure 1—figure supplement 3*, *Figure 2—figure supplement 1*). Furthermore, although some regions close to the DVC such as pons and the spinal cord show some co-expression of *Kcnj3* and *Gfap*, the patterns are not to the same extent as observed in the DVC (*Jessa et al., 2019*; *Figure 2—figure supplement 1*). In addition, we distinguished a spectrum of DVC mature oligodendrocytes, including a group resembling the intermediate oligodendrocytes described in the other brain sites that decrease in number with age (*Marques et al., 2016*; *Sathyamurthy et al., 2018*; *Figure 2C*). This gradient of expression of markers including *Fyn*, *Opalin*, and *Anln* in

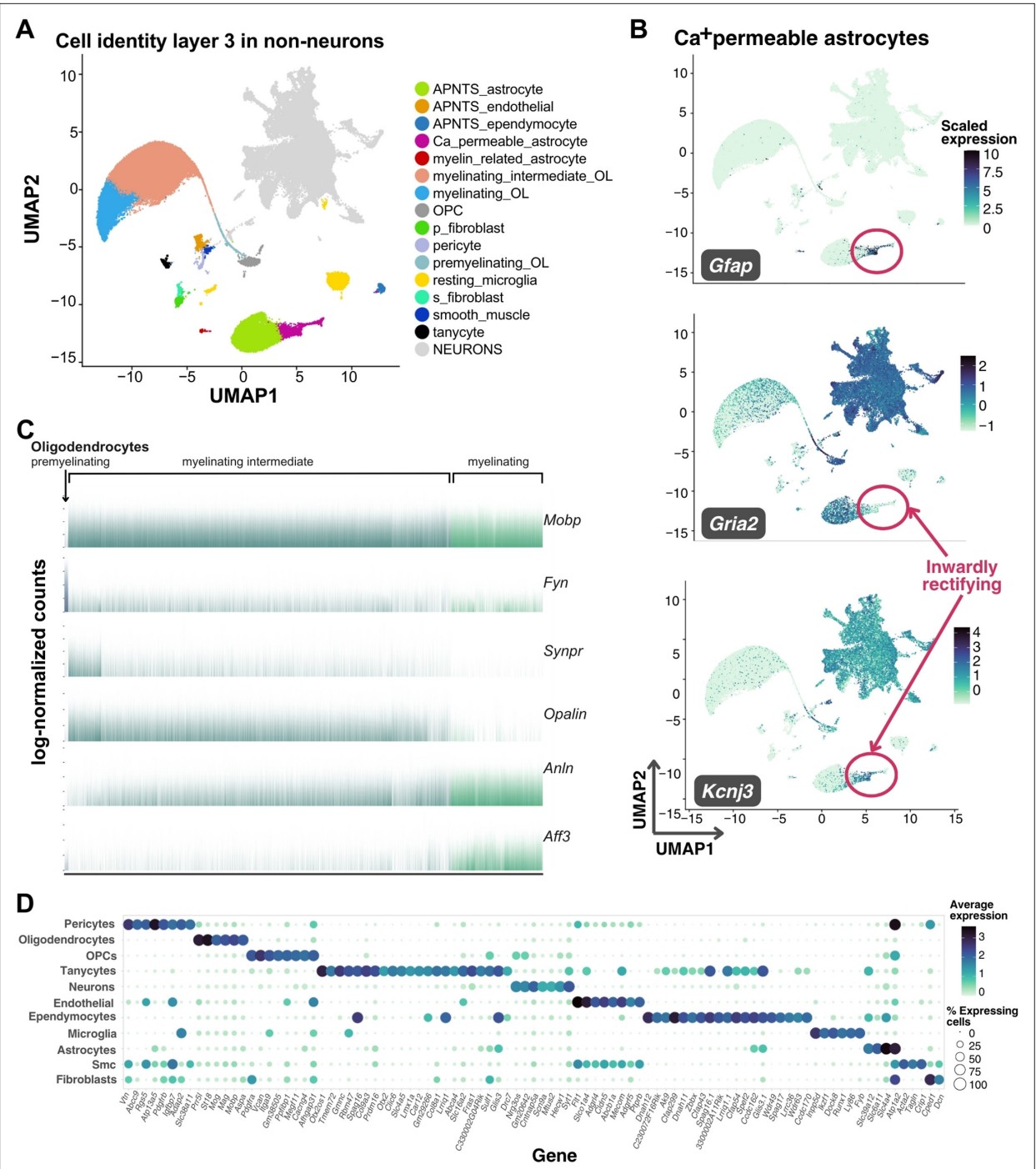

**Figure 2.** Third layer of non-neuronal murine DVC cell identities. (**A**) Labeled UMAP plot of the mouse DVC with the 15 non-neuronal cell identities shown (total cells $n$ = 99,740; non-neuronal cells $n$ = 51,609). (**B**) UMAP plot showing scaled expression of *Gfap*, *Gria2*, and *Kcnj3* genes in Ca$^+$-permeable astrocytes (circled; total cells $n$ = 99,740; Ca$^+$-permeable astrocytes $n$ = 1719). (**C**) Barplot of log-normalized expression of six genes in oligodendrocytes (premyelinating $n$ = 267; myelinating intermediate $n$ = 25,692; myelinating $n$ = 6170). Only premyelinating oligodendrocytes lack expression of *Mobp*, involved in myelin formation. Reduction of *Synpr* and *Opalin* expression is observed as *Anln* and *Aff3* increase in myelinating oligodendrocytes. (**D**) Balloon plot of the main cell types of non-neuronal cells and neurons showing average log-normalized counts of their marker genes (MAST algorithm; adj. p < 0.05). Markers shown are upregulated genes in >80% of cells per group with average log$_2$FC >4, or upregulated in >70% of cells with average log$_2$FC >8. DVC = dorsal vagal complex; UMAP = uniform manifold approximation and projection; APNTS = area postrema and nucleus of the solitary tract; OL = oligodendrocyte; OPC = oligodendrocyte precursor cell; Smc = smooth muscle cells.

The online version of this article includes the following figure supplement(s) for figure 2:

*Figure 2 continued on next page*

*Figure 2 continued*

**Figure supplement 1.** Mapping of DVC Ca⁺-permeable astrocytes in other murine brain sites.

**Figure supplement 2.** Trajectory inference of the oligodendrocyte lineage.

**Figure supplement 3.** Murine DVC clusters.

**Figure supplement 4.** Mouse microglial activation states.

oligodendrocytes (*Figure 2C*), does not respond to a developmental trajectory (*Figure 2—figure supplement 2*). Although 'myelinating-intermediate' and 'myelinating' oligodendrocytes include cells with overlapping gene expression programs, there is no evidence that each oligodendrocyte will follow a single trajectory (*Figure 2—figure supplement 2*). Furthermore, to properly separate some cell identities, we performed subclustering in three DVC clusters (i.e. clusters 23, 26, and 27) (*Figure 2—figure supplement 3*). This permitted us to label pre-myelinating oligodendrocytes, endothelial and mural cells, and distinguish two fibroblast subtypes (*Figure 2—figure supplement 3*). One of these subtypes expresses high *Stk39* previously described in fibroblast stress-response (*Kasai et al., 2022*) and so was termed 's-fibroblasts' for stress-associated fibroblasts (*Figure 2—figure supplement 3*), which we failed to detect in the hypothalamus (*Figure 1—figure supplement 3*). Additionally, we excluded the possibility of monocytes in our microglial data (*Figure 2—figure supplement 4*), which had no evidence of microglial activation as we mapped minimal levels of the activation genes *Cxcl10*, *Cd5*, *Cxcl9*, and *Zbp1* (*DePaula-Silva et al., 2019*) therefore , we posit that our DVC microglial cells are in a basal state (*Figure 2—figure supplement 4*). To further evaluate possible activation states in microglia, we performed subclustering and evaluated their marker genes (*Figure 2—figure supplement 4*). We have identified a group of microglia expressing oligodendro-cyte markers and resembling a subpopulation of hippocampal microglia in an Alzheimer's disease model which is disease-protective and synaptic-function-supporting (*Koutsodendris et al., 2023*), and a cortical/subcortical phagocytic subpopulation in a multiple sclerosis model (*Schirmer et al., 2019*). A second group expresses *Ndrg2* and receptors for GABA and glutamate which have been related to microglia that may induce neuronal damage (*Creus-Muncunill et al., 2024*; *Tang et al., 2016*; *Figure 2—figure supplement 4*).

We also obtained the DVC gene expression markers of the main cell types in the central nervous system and neurons (*Figure 2D*). To more thoroughly classify DVC neural types, we subset and re-clustered the 10 neural classes from layer-2, which resulted in 35 neuronal identities as a third layer of granularity (*Figure 3A*; *Supplementary file 6*). These identities include 'preganglionic' or *Chat*-expressing neurons and a large number of neurons with unspecific markers which we called 'mixed neurons' (*Supplementary file 6*). Of the mixed neuron classes, two are largely excitatory (mixed neurons 3 and 4) and two largely inhibitory; however, cells in the four mixed neuron classes share similar transcriptional profiles with many other neural classes. When we performed subclustering on mixed neurons, we could distinguish a total of 10 subtypes from the original 4 mixed neuronal identities (*Figure 3—figure supplement 1*); however, transcriptional differences between these were subtle. Interestingly, a mixed neurons-like group of cells was previously described in the murine hypothalamus (*Campbell et al., 2017*), suggesting multiple brain sites may contain neurons which lack highly differentiable transcriptional programs from snRNA-sequencing data.

Additionally, we split the monoamine class into four layer-3 cell identities (i.e. M0, M1, M2, and M3) with specific cell markers (*Figure 3B*; *Supplementary file 6*). M0 and M1 neurons express genes for enzymes synthesizing norepinephrine and serotonin, but M0 expresses minimal levels of the transporter VMAT2 (*Slc18a2*) (*Figure 3C*; *Supplementary file 6*). Due to the low proportion of cells expressing *Th* and *Tph2* in M2 and M3 neurons, we consider them monoamine modulators as they seem to synthesize trace amines (*Figure 3C*; *Supplementary file 6*). Of note, the receptor for GDF15, GFRAL, is expressed heavily by a subset of M0 neurons, some of which co-express the calcium sensing receptor (*Casr*) gene (*Figure 3C*).

We named 'GLP1' a neuronal cluster with high expression of the pre-proglucagon gene (*Gcg*) (*Figure 3D*; *Supplementary file 6*), which also expresses high levels of both the prolactin and leptin receptors (*Cheng et al., 2020b*; *Figure 3E*). Additionally, these cells have the highest proportion of DVC neuronal co-expression of glutamate- and GABA-related genes (*Figure 3F*); mainly *Gad2* with considerably lower expression of *Slc32a1*.

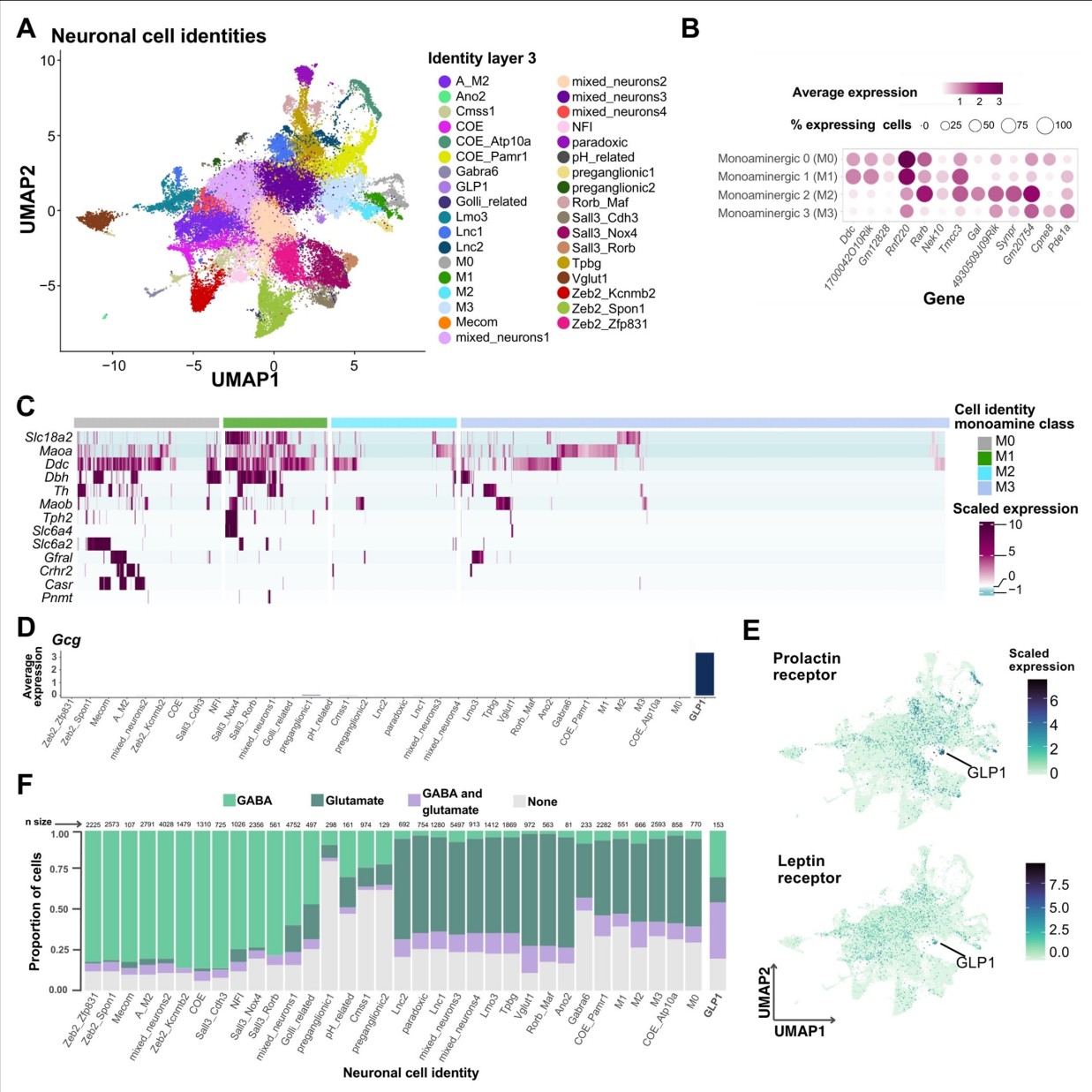

**Figure 3.** Neuronal populations of the murine DVC. (**A**) Labeled UMAP plot with the neuronal layer-3 cell identities (neurons n = 48,131). (**B**) Monoamine class balloon plot showing average log-normalized expression of marker genes for each cell identity (MAST algorithm; M0 cells n = 770, M1 cells n = 551, M2 cells n = 666, and M3 cells n = 2593; adj. p < 0.05), and (**C**) heatmap of expression of monoamine-related genes. In the heatmap, each column represents one cell and each row, one gene. (**D**) Barplot of *Gcg* average log-normalized expression by neuronal cell identity. (**E**) UMAP plot showing the expression of leptin receptor and prolactin receptor genes in neurons. The GLP1 cluster is highlighted (GLP1 cells n = 153). (**F**) Stacked bar plot of the proportion of cells expressing one or more GABA-related genes (i.e. *Slc32a1*, *Gad1*, *Gad2*) and glutamate-related genes (i.e. *Slc17a6*, *Slc17a7*). The proportion of cells per cell group co-expressing GABA and glutamate associated genes is shown in purple. Only cells with log-normalized counts >0 were considered to express the genes. DVC = dorsal vagal complex; UMAP = uniform manifold approximation and projection; Res. = resolution; GABA = gamma-aminobutyric acid.

The online version of this article includes the following figure supplement(s) for figure 3:

**Figure supplement 1.** Mixed neurons subclustering.

In our analysis, we found a subset of monoaminergic neurons that express the cholecystokinin (*Cck*) gene (**Figure 4A**). While the *Cck*- and *Th*-expressing DVC neurons are considered to be non-overlapping cell types with distinct physiologies (**Roman et al., 2016**; **Roman et al., 2017**), we quantified >10% of the *Th*-expressing neurons also expressing *Cck* in our snRNA-seq data. We further

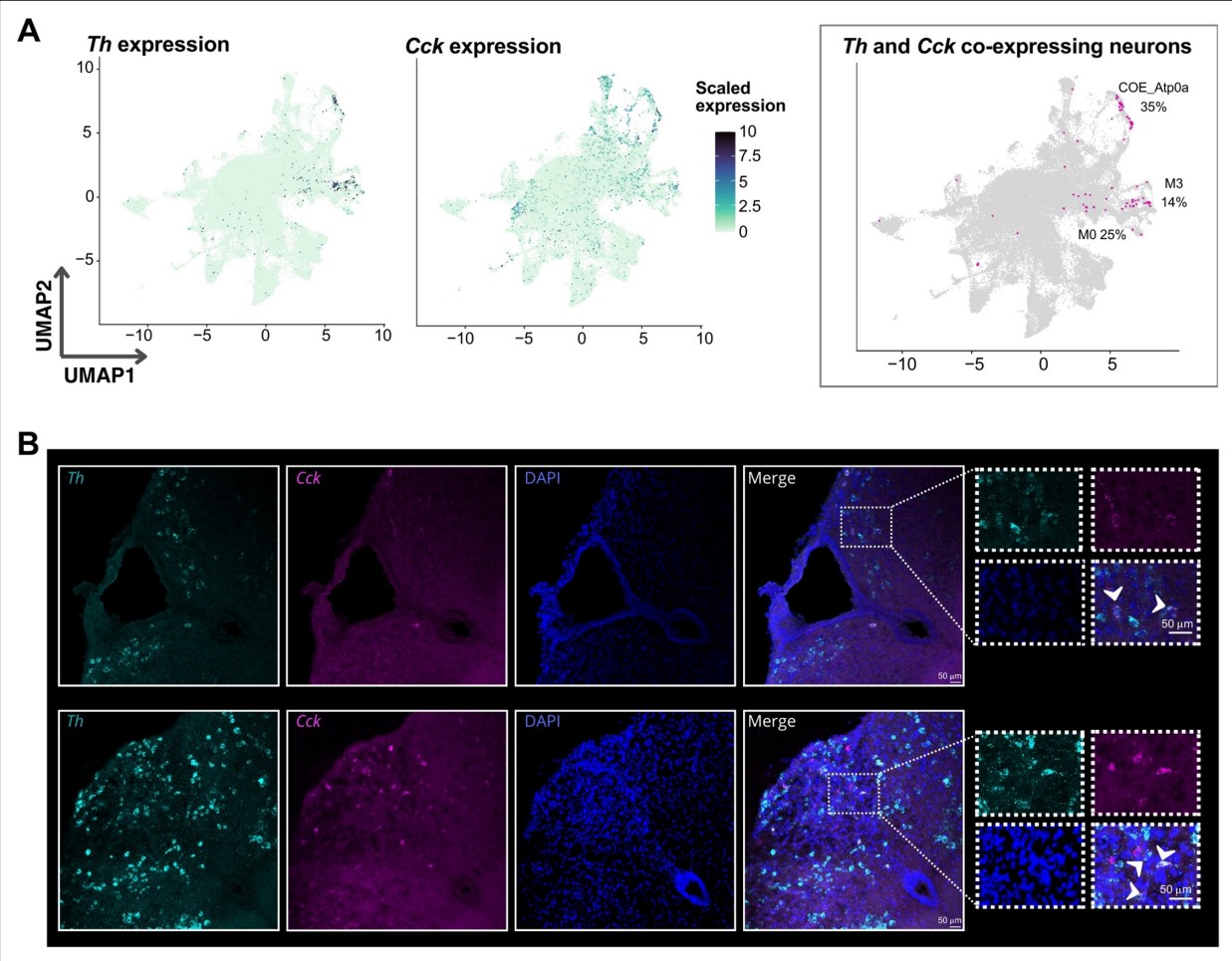

**Figure 4.** *Th* and *Cck* co-expression in the DVC. (**A**) UMAP plot of neuronal *Th* and *Cck* scaled expression and cells with co-expression of both genes (neurons n = 48,131; *Th*-expressing neurons n = 764; *Cck*-expressing neurons n = 3821; *Th*/*Cck* co-expressing neurons n = 80). On the right plot, cells highlighted in magenta are the co-expressing neurons. The three neuronal cell identities in which the majority of nuclear co-expression of *Th* and *Cck* mRNA is found, as well as the percentage of total co-expressing neurons, are shown. (**B**) In situ hybridization of *Cck* and *Th* mRNA in coronal mouse DVC sections corresponding to –7.2 and –7.56 mm relative to bregma showing overlap of *Th*/*Cck*. Some of the cells with overlapping signal are highlighted with an arrow in the enhanced merged image. DVC = dorsal vagal complex; UMAP = uniform manifold approximation and projection.

confirmed co-expression of *Th* and *Cck* results by in situ hybridization (***Figure 4B***). The majority of this overlap occurs in cells belonging to the COE-Atp10a and monoamine class (i.e. M0, M3) cell identities (***Figure 4A***).

## The murine DVC cell atlas

To generate a comprehensive murine DVC database from multiple datasets, we utilized treeArches (***Michielsen et al., 2023***) to harmonize our snRNA-seq DVC labels with that from a publication by Ludwig and collaborators (i.e. the 'Ludwig dataset') (***Ludwig et al., 2021***; ***Figure 5A, B***). After building our initial cell hierarchy (***Figure 5—figure supplement 1***), we integrated our dataset with the Ludwig dataset, giving a total of 171,868 cells (***Figure 5B***). Next, the labeled cell identities from the Ludwig dataset (***Ludwig et al., 2021***) were incorporated into our tree based on the similarity between the transcriptomic programs of our labeled identities and the Ludwig-labeled identities using progressive learning through scHPL (***Michielsen et al., 2021***; ***Figure 5A***). Some of the Ludwig dataset identities failed to be incorporated into our tree, like the oligodendrocyte precursor cells (OPCs), which contain a mix of OPCs and pre-myelinating oligodendrocytes (***Figure 5—figure supplement 2***). The mix of cell programs makes it impossible for treeArches to correctly add one branch of the Ludwig OPCs to the tree, as ideally, it would be added as a subpopulation of both the OPCs and pre-myelinating

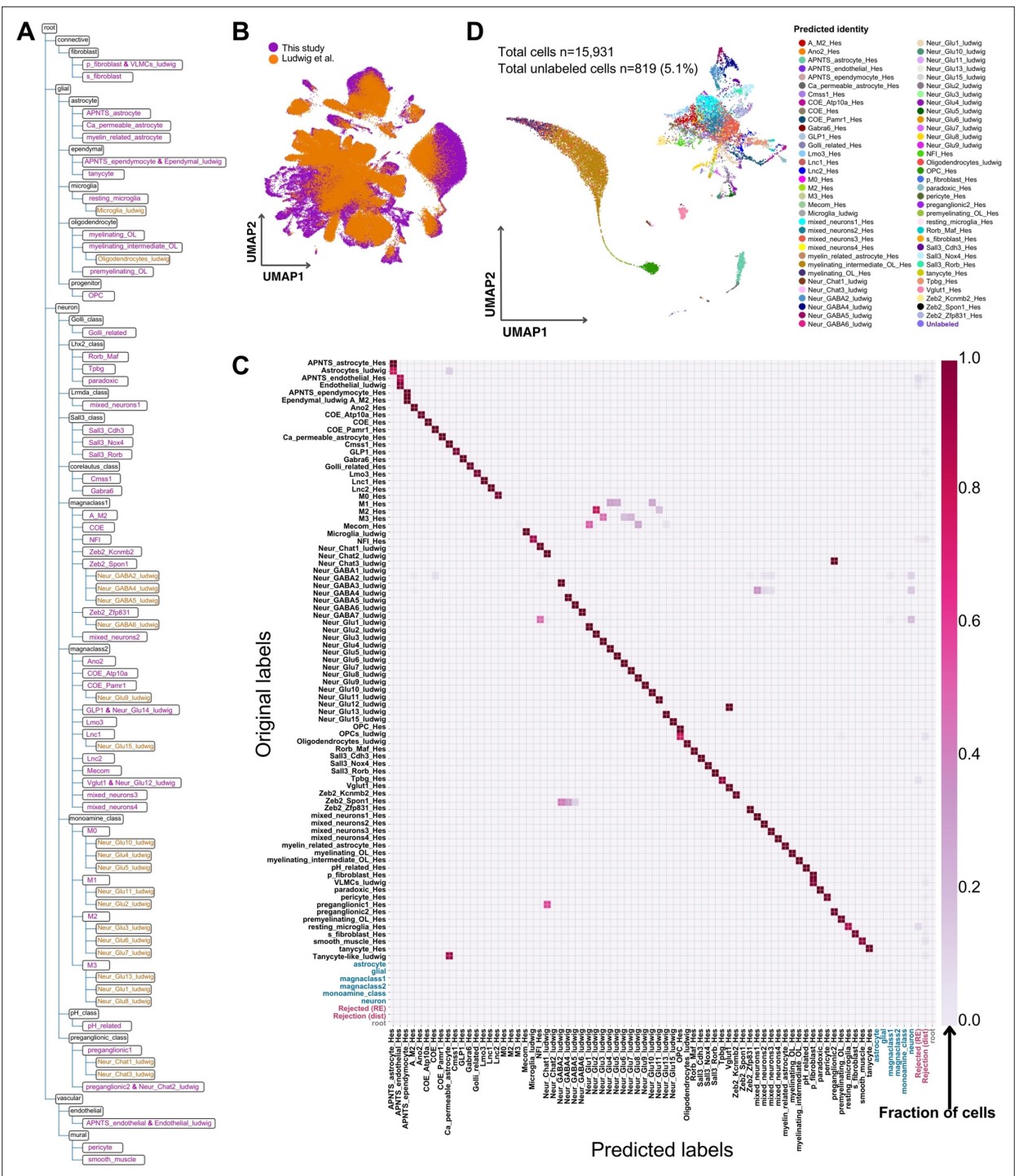

**Figure 5.** The murine dorsal vagal complex (DVC) cell hierarchy. (**A**) Dendrogram of the harmonized hierarchy which incorporates cells from Ludwig and our datasets. Orange cell identities are a fourth layer of cell identity resolution obtained as some of the Ludwig dataset identities are subgroups of our original cell identities at their highest resolution. Magenta labels represent layer-3 of cellular granularity. These two layers are considered high resolution. (**B**) UMAP plot of the integration between our and Ludwig datasets using treeArches (cells *n* = 171,868). (**C**) Pairwise heatmap showing the proportion of cells originally labeled in this study and in Ludwig dataset (*y*-axis), predicted to belong to each identity group (*x*-axis) by treeArches using our learned harmonized hierarchy. In blue are the labels considered non-specific for a high-resolution cell identity (*n* = 2324; 1.3%). In pink are the cell labels for rejected cells, therefore not assigned any identity by treeArches (*n* = 3108; 1.8%). (**D**) UMAP plot of the Dowsett dataset labeled through treeArches using the learned harmonized hierarchy representation from our murine DVC atlas. The 'unlabeled' cells include those rejected by the algorithm and thus without an assigned identity, and those with an unspecific layer-3/4 label. For example, some were labeled 'neuron' or 'monoamine

*Figure 5 continued on next page*

*Figure 5 continued*

class' but without a high-resolution cell identity. UMAP = uniform manifold approximation and projection; APNTS = area postrema and nucleus of the solitary tract; Lnc = long non-coding; OL = oligodendrocyte; OPC = oligodendrocyte precursor cell; Neur = neuronal; COE = collier/Olf1/EBF transcription factor.

The online version of this article includes the following figure supplement(s) for figure 5:

**Figure supplement 1.** Initial murine cell hierarchy using our samples.

**Figure supplement 2.** Incorporation of the Ludwig dataset to our initial hierarchy using treeArches.

oligodendrocytes, which is not biologically possible. The resultant murine hierarchy has a fourth layer of cellular resolution as some of Ludwig identities, mainly neurons, are subgroups of our cell identities (*Figure 5—figure supplement 2*).

We confirmed the validity of our cell hierarchy using three methods. First, by using the learned model to predict the labels of the cells we used to develop them (i.e. this study and the Ludwig dataset), for which we obtained highly concordant cell labeling (*Figure 5C*). Secondly, we used the learned model to predict the labels of another publicly available dataset from Dowsett and collaborators (i.e. the 'Dowsett dataset') (*Dowsett et al., 2021*; *Figure 5D*). Using this approach, we were able to successfully label 95% of these cells at high resolution (i.e. layer-3 or -4 of cell identity granularity); further validating our combined DVC labels. Finally, we manually mapped the marker genes from

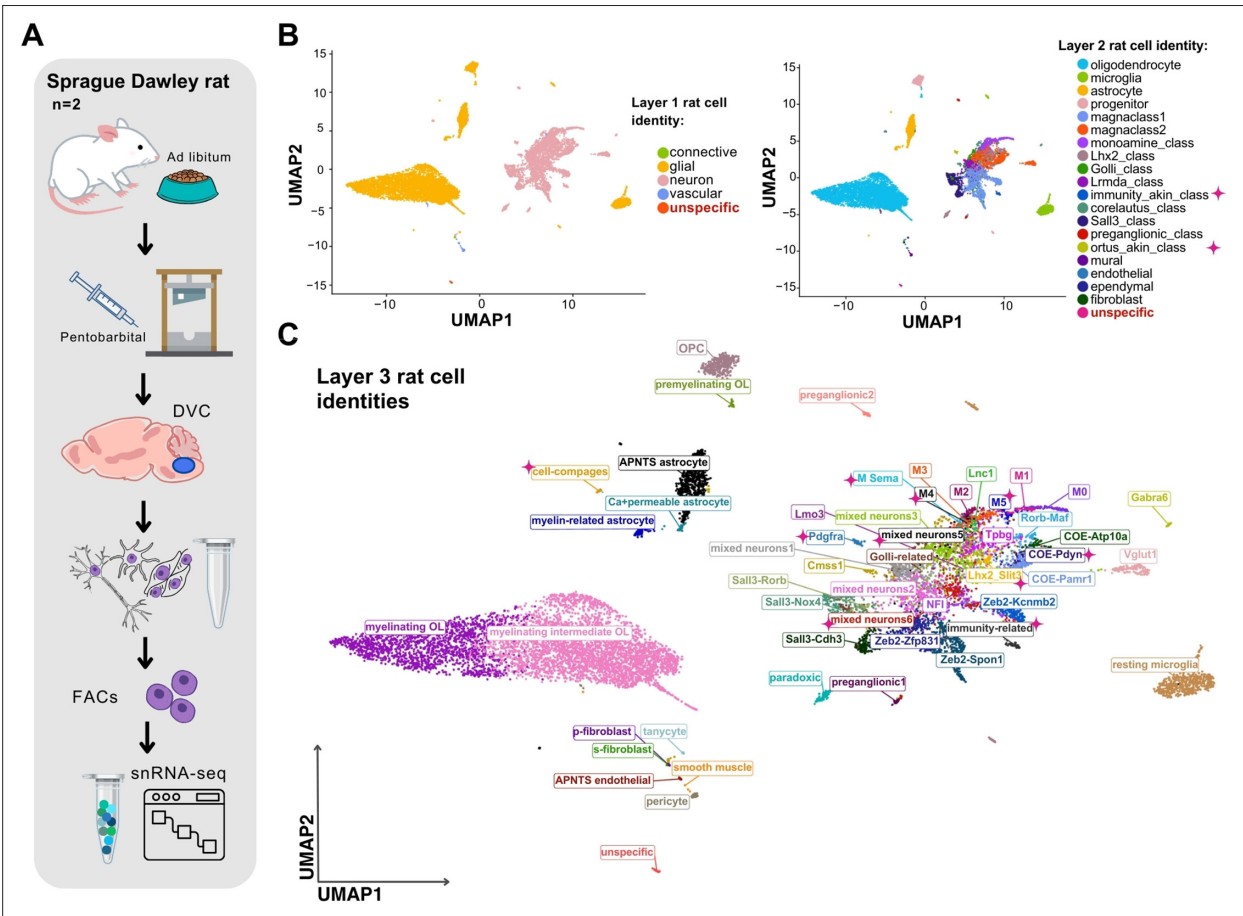

**Figure 6.** The snRNA-seq-derived cell identities for the rat DVC. (**A**) Schematic of the pipeline for snRNA-seq of the rat DVC. (**B**) Labeled UMAP plots of the low resolution (i.e. layers 1 and 2 of granularity) and (**C**) high resolution (i.e. layer-3) cell identities in our rat DVC dataset (cells *n* = 12,167). We labeled 4 layer-1, 19 layer-2, and 52 layer-3 cell identities. Those labels novel to this dataset and not present in the murine data are highlighted with a pink ♦ symbol. We found a small cluster (cells *n* = 35) that we could not corroborate its identity at any layer of cellular granularity that we named 'unspecific'. snRNA-seq = single-nuclei RNA-sequencing; DVC = dorsal vagal complex; FACS = fluorescence-activated cell sorting; UMAP = uniform manifold approximation and projection; APNTS = area postrema and nucleus of the solitary tract; Lnc = long non-coding; OL = oligodendrocyte; OPC = cursor cell; Neur = neuronal; COE = collier/Olf1/EBF transcription factors.

Ludwig (*Ludwig et al., 2021*) and Dowsett (*Dowsett et al., 2021*) neuronal clusters in our dataset and confirmed the subgroups of neurons among our identities pointed out by treeArches (*Figure 5—figure supplement 2*).

## An integrated rodent DVC cell hierarchy

To identify rat DVC cell clusters and subclusters and contrast these with the mouse DVC atlas, we generated a de novo ad libitum-fed rat dataset (cells *n* = 12,167) (*Figure 6A*) and labeled it by mapping the top 5 gene expression markers of each of our murine layer-3 cell identities (*Supplementary file 7*). After manual curation at three granularities (*Figure 6B, C*; *Supplementary file 3*), we obtained 52 DVC rat identities at high cellular resolution (*Figure 6C*) including a small group of 'unspecific' cells (*n* = 35) that we could not confidently label because their top gene expression markers have unknown functions and none of our known identities' genes clearly marked this population (*Figure 6B, C*; *Supplementary file 7*).

We established 10 layer-3 neuronal identities not previously found in the murine data (*Figure 6C*; *Figure 7—figure supplement 1A*; *Supplementary file 3*). Notably, two of those did not belong to any of the existing murine neuronal classes (i.e. layer-2 of cellular granularity); therefore, we created two new layer-2 classes (*Figure 6B*). One of these, the immunity-related neurons, has high *Csf1r* expression, as well as cystatin C and *Inpp5d* which are commonly associated with immunity-related processes in microglia and with immunomodulation in cancer (*Chou et al., 2023*; *Kleeman et al., 2023*; *Figure 7*; *Supplementary file 8*). The second neural class identified in rat but not mouse DVC was the 'Pdgfra neurons' (*Figure 7*). These neurons express *Pdgfra*, *Arhgap31*, and *Itga9*, known markers for progenitor cells for which we named this cell class 'ortus-akin' (Latin: 'origin') because they are similar to cells giving origin to other cells (*Kang et al., 2010*; *Figure 7A*; *Supplementary file 8*). Interestingly, this neuronal class not only expressed *Pdgfra*, but also other OPC signature genes (*Figure 7A*). Mapping the expression levels of metabolic-associated receptors revealed that a subset (~15%) of both immunity-akin and ortus-akin express detectable *Gfral* mRNA, but this is likely an underrepresentation due to the challenges in detecting lowly expressed transcripts such as *Gfral*. Prolactin receptor and leptin receptor (*Figure 7B*) were also found to be expressed by these neurons, although they seem to have a varied neurotransmitter profile (*Figure 7—figure supplement 1*). To confirm the presence of *Pdgfra*-expressing neurons in the rat DVC, we performed immunostaining in the mouse and rat brainstem. Within the mouse, we found PDGFRA-immunoreactivity (IR) in the two morphologically distinct cells; matching OPC and blood vessel-associated fibroblast morphologies (*Figure 7C*). However, within the rat, we identified PDGFRA-IR in cells with three morphologies, one of which resembled neurons with an enlarged cell body and protruding cellular processes (*Figure 7C*). To determine if these PDGFRA-IR cells were neurons, we co-stained with the neural marker, HuC/D, and noted several PDGFRA and HuC/D copositive cells (*Figure 7D*), confirming our sequencing data that a PDGFRA-expressing neural class exists in the rat DVC and localizing these cells to the AP.

Other differences found between mouse and rat DVC expression include increased expression of the leptin receptor in neurons as well as cholecystokinin (*Figure 7—figure supplement 2*).

To construct an integrated rodent DVC cell atlas, we incorporated the labeled rat dataset into our hierarchy using treeArches (*Michielsen et al., 2023*; *Figure 8*). The rat data was projected onto the mouse datasets (*Figure 8A*) and the 42 high-resolution rat cell identities were appended to the tree (*Figure 8B*; *Figure 8—figure supplement 1*). The final hierarchy has 123 labels, of which 99 are high resolution according to our cellular granularity (i.e. layers 3–5) (*Figure 8B*). In addition, it has 20 layer-2 identities and 7 cell identities that are novel from the rat dataset (*Figure 8B*).

## *Hcrt* and *Agrp* expression in the DVC

Since *Agrp*-expressing cells have been recently described within the DVC (*Bachor et al., 2024*), we mapped this and other neuropeptide genes in DVC neurons. Surprisingly, we found *Hcrt* and *Agrp* mRNA in neurons in our dataset (*Figure 8—figure supplement 2*). We then performed in-situ hybridizations for these transcripts in both the mouse hypothalamus and DVC. While both *Hcrt* and *Agrp* were readily detectable in the hypothalamus, we failed to observe signal within the DVC (*Figure 8—figure supplement 2*). Since we also found expression of *Hcrt* and *Agrp* in neurons of the Dowsett and Ludwig datasets (*Ludwig et al., 2021*; *Dowsett et al., 2021*; *Figure 8—figure supplement 2*), we then mapped the sequenced reads from these transcripts and confirmed that most of them mapped

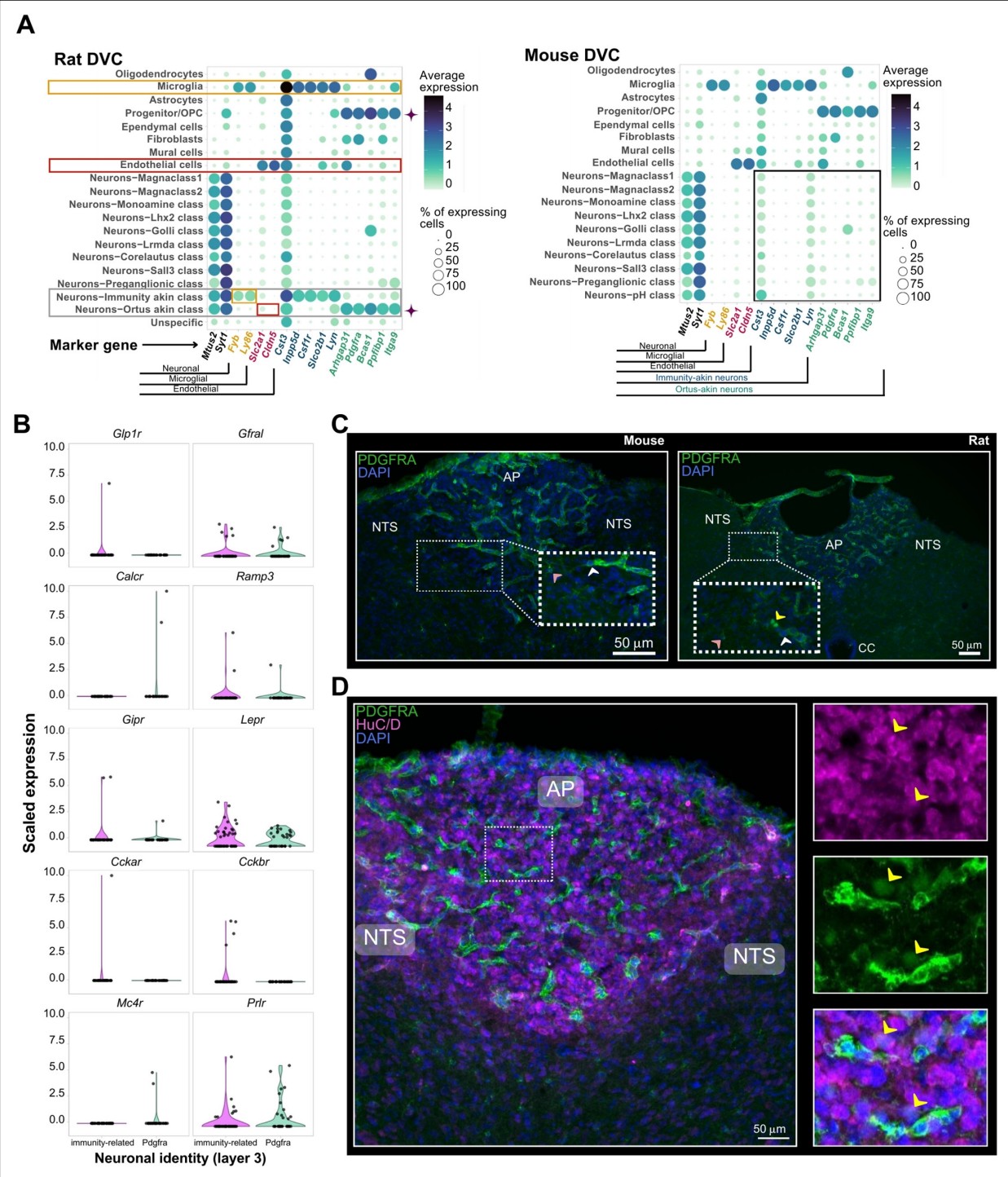

**Figure 7.** Two novel neuronal classes specific to the rat DVC. (**A**) Balloon plots comparing the expression of the marker genes of the two novel neuronal classes found in rats (cells *n* = 12,167): immunity-akin and the ortus-akin classes (framed in gray). Expression is shown across the layer-2 rat cell identities. Based on the expression of two DVC neuronal markers *Mtus2* and *Syt1*, we corroborated those cell classes contain neurons, and not microglial or vascular/endothelial cells (In orange and red, respectively). The immunity-akin class (*n* = 48 neurons) shares high expression of *Cst3*, *Inpp5d*, *Csf1r*, *Slco2b1*, and *Lyn* with microglial cells. Meanwhile, the ortus-akin class (*n* = 34 neurons) shows higher overlap of gene expression markers (i.e. *Arhgap31*, *Pdgfra*, *Bcas1*, *Ppfibp1*, and *Itga9*) with OPCs (highlighted with a ♦ symbol). Mouse cell identities (cells *n* = 99,740) do not show these expression patterns (framed in black). Average expression was calculated using log-normalized counts. (**B**) Violin plots of scaled expression of 10 genes (i.e. *Glp1r*, *Gfral*, *Calcr*, *Ramp3*, *Gipr*, *Lepr*, *Cckar*, *Cckbr*, *Mc4r*, and *Prlr*) coding for metabolism-associated receptors in the Pdgfra and immunity-related rat neurons (Pdgfra neurons *n* = 34; immunity-related neurons *n* = 48). Pdgfra neurons are the only layer-3 identity within the ortus-akin class. An

*Figure 7 continued on next page*

*Figure 7 continued*

overlapping dot plot shows one dot per cell. (**C**) Immunofluorescent detection of PDGFRA in mouse and rat DVC. Pink arrowheads point to cells with OPC morphology, white arrowheads point to cells with fibroblast morphology, and the yellow arrowhead points to cells with neural morphology. (**D**) Co-staining for PDGFRA and HuC/D in rat DVC with high magnification (right images) of area postrema PDGFRA-expressing cells. Yellow arrowheads indicate colocalization of HuC/D and PDGFRA. DVC = dorsal vagal complex; OPC = oligodendrocyte precursor cell; AP = area postrema; NTS = nucleus of the solitary tract; PDGFRA = platelet-derived growth factor receptor A; CC = central canal.

The online version of this article includes the following figure supplement(s) for figure 7:

**Figure supplement 1.** Rat DVC neurons.

**Figure supplement 2.** Receptors and neuropeptide expression in mouse and rat neurons.

to intronic and non-coding regions of these genes (*Figure 8—figure supplement 2*). Furthermore, our rat DVC dataset was pre-processed excluding introns from the reads mapping, and the *Hcrt* and *Agrp* expression levels observed are minimal (*Figure 8—figure supplement 2*), which is in agreement with the non-coding location of the reads in mouse DVC. These data explain the absence of signal from our in situ hybridizations and suggest DVC cells do not produce HCRT or AGRP neuropeptides.

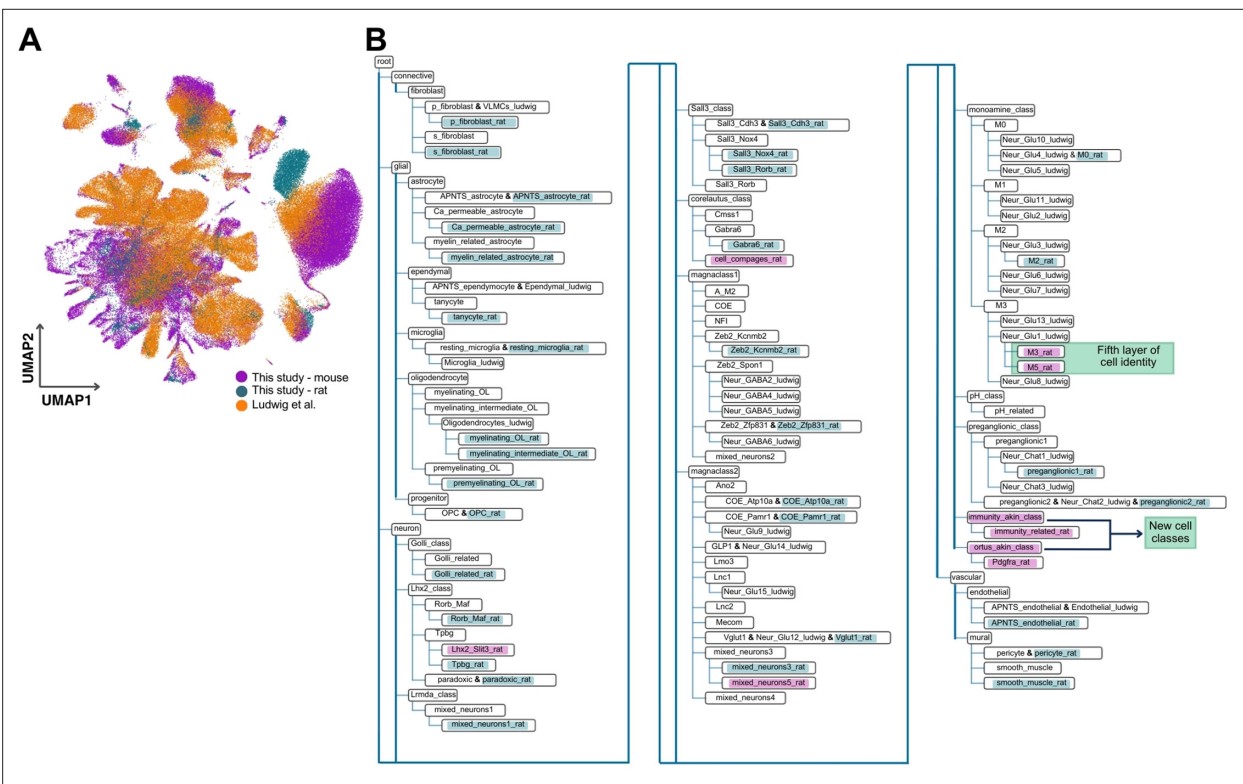

**Figure 8.** The rodent DVC cell hierarchy. (**A**) UMAP plot of the integration between the murine hierarchy and our rat dataset using treeArches (cells *n* = 184,035). (**B**) Dendrogram of the harmonized hierarchy of our labeled rat dataset and the murine DVC hierarchy. We highlight the incorporated rat cell identities (blue and magenta). Those in magenta are the novel rat identities established by us. The immunity-akin and the ortus-akin classes were not found in the murine datasets. The M3 and M5 rat identities are subgroups of a Ludwig layer-4 cell identity, therefore yielding a fifth layer within the monoamine neuronal class. UMAP = uniform manifold approximation and projection; DVC = dorsal vagal complex; APNTS = area postrema and nucleus of the solitary tract; Lnc = long non-coding; OL = oligodendrocyte; OPC = oligodendrocyte precursor cell; Neur = neuronal; COE = collier/Olf1/EBF transcription factors.

The online version of this article includes the following figure supplement(s) for figure 8:

**Figure supplement 1.** Dendrogram after incorporation of the rat dataset to the murine hierarchy.

**Figure supplement 2.** Expression of *Agrp* and *Hcrt* in DVC neurons.

## Meal-related transcriptional programs in the murine DVC

As the DVC acts as a primary node for meal-related signals (*Holt, 2022*), we wanted to define which of our cell identities responded to meal consumption and thus performed differential gene expression analysis between DVC cell types from mice that were euthanized under fasting conditions, with ad libitum access to food or 2 hr post-refeeding following an overnight fast (*Figures 1A and 9*). This analysis revealed widespread transcriptional changes across nearly every neural cell identity, the exceptions being corelautus, Golli-class, and pH classes (*Figure 9A*). Additionally, the size of the transcriptional changes was relatively small, with approximately 90% of the differentially expressed genes having increases or decreases between 0.5 and 1.0 $\log_2$ fold-change (FC), with only ~10% of the gene changes being ≥1 $\log_2$FC between conditions (*Figure 9A*). Of the responding neurons, magnaclasses 1 and 2, which represent large populations of predominately inhibitory and excitatory neurons (*Figure 1—figure supplement 1*), showed the greatest number of differentially expressed genes following refeeding for both comparisons refed versus fasting and refed versus ad libitum food access (*Figure 9A*). To understand how refeeding transcriptionally alters inhibitory compared with excitatory DVC neurons, we compared differentially expressed genes between these two classes across conditions. Intriguingly, despite being in transcriptionally distinct classes, we found meal consumption altered many of the same genes (*Figure 9B–D*), suggesting these cells may be receiving similar postprandial signals which are propagated via these cells in unique manners. This is supported by the similar receptor expression profile of magnaclasses 1 and 2 (*Figure 9—figure supplement 1*). One notable exception is *Cckbr*, which is upregulated in magnaclass 2 and downregulated in magnaclass 1 in refeeding when compared with ad libitum fed mice (*Supplementary file 9*). We also note that oligodendrocytes contained the highest number of differentially expressed genes in refeeding (*Figure 9A*). While the roles of oligodendrocytes in refeeding responses within the DVC have yet to be determined and the biologic significance of these changes is presently unclear, DVC oligodendrocytes also have robust transcriptional responses in response to fasting (*Dowsett et al., 2021*), suggesting these cells are sensitive to both positive and negative energy balance associated cues and may have important roles in regulating energy homeostasis.

## Discussion

The recent success of DVC-based therapies (*Véniant et al., 2024*; *Breen et al., 2020*) has moved DVC biology to the forefront of weight control therapies and there is now clear clinical impetus to fully define DVC cell types and their functions. Previous studies detailing DVC cell types describe a high degree of heterogeneity (*Ludwig et al., 2021*; *Dowsett et al., 2021*; *Zhang et al., 2021*); however, these studies lacked an in-depth analysis of neurotransmitter profiles of neurons, any cross-species evaluation of DVC cell types, or an analysis of how meal consumption transcriptionally regulates these cells. We therefore developed de novo mouse and rat DVC cell atlases to address these points.

To more fully describe the cellular complexity within the DVC, we combined a pipeline of cell label transfer using brain existing published datasets and thorough curation of cell identities, which resulted in a four-layer mouse cell hierarchy and a five-layer rodent cell hierarchy of the DVC. Within this hierarchy, we found several unique features of DVC glial cells. First, we found Ca+-permeable astrocytes in the DVC in which *Gfap* is almost exclusively expressed and they co-express *Kcnj3* (GIRK1 potassium channel). These cells largely lack *Gria2*, consistent with findings in Bergmann glial cells in which colocalization of *Gria2* and GFAP was found to alter neuron-glial interactions (*Tsuzuki and Ishiuchi, 2008*). The heterogeneity of astrocytes across, and even within, spatially distinct CNS sites is becoming increasingly apparent (*Endo et al., 2022*), and our finding of DVC-specific astrocyte transcriptional profiles further supports this. When we compared Ca+-permeable DVC astrocytes to hypothalamic astrocytes (*Steuernagel et al., 2022*), we found the same expression pattern with minimal overlap between *Gria2* and *Gfap*. However, DVC astrocytes seem to have some unique properties when compared with hypothalamic, cortical, and hippocampal astrocytes. Principally, DVC astrocytes express the GIRK1 channel, likely rendering them less excitable (*Luján, 2010*). As astrocytes have well-established roles in maintaining the blood–brain barrier function (*Cabezas et al., 2014*), and the pattern of GFAP in the DVC is largely restricted to the boundary separating the AP and NTS, it is possible these DVC-specific astrocytes have specialized functions in regulating blood–brain barrier permeability.

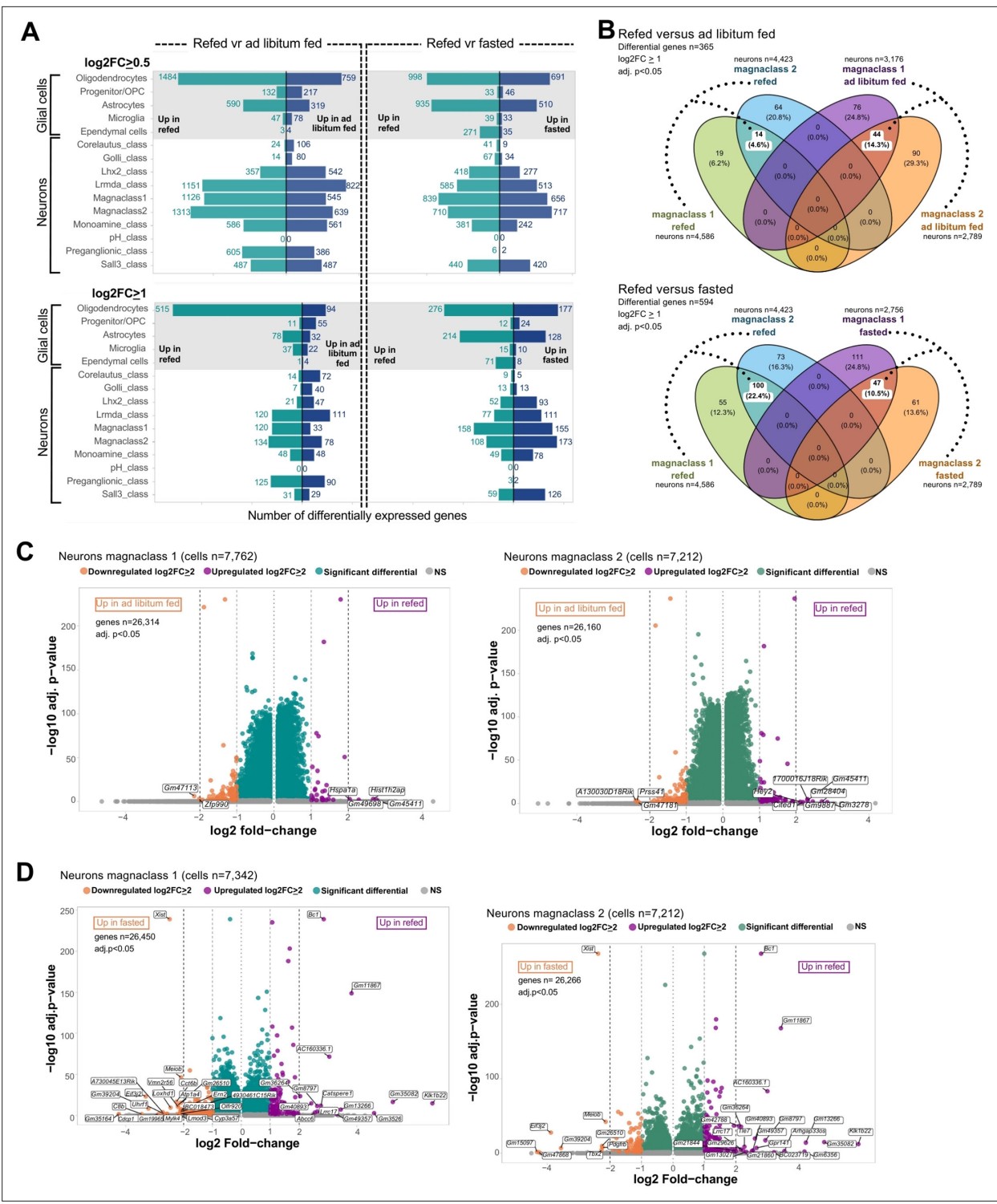

**Figure 9.** Meal-related transcriptional changes in the mouse DVC. (**A**) Horizontal bar plots of the number of differential genes between refed and ad libitum fed, and refed and fasted mice in glial cells and neurons per layer-2 cell identity (MAST algorithm on log-normalized counts; adj. p < 0.05; refed neurons *n* = 14,396; refed glial cells *n* = 18,019; ad libitum fed neurons *n* = 9885; ad libitum fed glial cells *n* = 4610; fasted neurons *n* = 8829; fasted glial cells *n* = 12,496). (**B**) Venn diagrams of the differentially expressed genes in magnaclass 1 and 2 neurons between refed and ad libitum fed, and between refed and fasted treatments (MAST algorithm on log-normalized counts). The number of overlapping log$_2$FC ≥1 upregulated genes between the two magnaclasses per treatment is highlighted. The treatment-induced changes in only one cell class are shown as non-overlapping. The percentage is based on the total differential genes surveyed per comparison. (**C**) Volcano plots of the differential genes in neurons belonging to the magnaclasses 1 and 2 between refed and ad libitum fed, and (**D**) refed and fasted treatments (MAST algorithm on log-normalized counts; adj. p < 0.05). Each point

*Figure 9 continued*

represents one gene. Only genes upregulated or downregulated with a log$_2$FC ≥2 are labeled. Since we considered a minimum log fold-change of 0.1 between treatments per cell group, those genes with very low variance (i.e. log fold-change <0.1) were excluded from the differential expression analysis; therefore, a variable number of genes are shown per comparison per identity in the volcano plots. DVC = dorsal vagal complex; vr = versus; OPC = oligodendrocyte precursor cell; FC = fold-change; adj. = adjusted; NS = non-significant.

The online version of this article includes the following figure supplement(s) for figure 9:

**Figure supplement 1.** Expression of metabolism-associated receptors in the DVC magnaclasses.

Additionally, while we identified hypothalamic fibroblasts that express high *Aldh1a1* similar to the DVC p-fibroblasts, we failed to find a group equivalent to s-fibroblasts. Other cell populations seem to be consistent with what has been found in other brain sites, like the intermediate oligodendrocytes that are found enriched in the hypothalamus and the corpus callosum in mice (*Marques et al., 2016*). In summary, these findings show that while certain DVC glial cells share identities with glial cells from other CNS sites, there are also DVC-specific glial cells.

Within the mouse neural classes, we found overlap between *Cck-* and *Th*-expressing cells. As these cells have previously been described as non-overlapping, this was unexpected (*Roman et al., 2016*). Our in situ hybridization confirmed the presence of *Cck/Th* positive cells within the NTS and AP. We believe the discrepancy may be due to detection methods as these cells were initially defined as non-overlapping using immunofluorescent detection of TH protein and a *Cck* genetic reporter (*Roman et al., 2016*). Functional studies of *Cck-* and *Th*-expressing neurons within the NTS demonstrate these cells have somewhat distinct roles; while both reduce appetite, *Cck* neurons are strongly aversive whereas *Th*-labeled neurons are non-aversive (*Cheng et al., 2020a*). While no functional characterization of *Cck* and *Th* neurons of the AP has been performed, we posit that as some *Th*-expressing AP neurons also express *Gfral* and *Casr*; both having potently aversive ligands (e.g. GDF15, deoxynivalenol) (*Yang et al., 2017*; *Patel et al., 2023*); they are likely to mediate aversive anorexia. The role of dual *Cck/Th* positive neurons remains to be determined.

While AGRP expression has long been considered to be hypothalamic-exclusive, a recent report describes *Agrp*-expressing cells within the mouse DVC (*Bachor et al., 2024*), in part from the detection of Agrp mRNA in scRNA-seq data. Similarly, we also detected transcripts for both *Agrp* and *Hcrt* in our snRNA-seq data but failed to detect any transcripts within the DVC by in situ hybridizations. One possible explanation for the discrepancy between the snRNA-seq data and our in situ hybridizations data is that the snRNA-seq mRNA reads for *Agrp* and *Hcrt* gene overwhelmingly mapped to non-coding regions of the respective genes. Combined, these data suggest to us that neither AGRP nor HCRT neuropeptides are produced by DVC cells in rodents.

We show the presence of neurons with seemingly mixed programs in the DVC, which we called 'mixed neurons'. Similar groups have been described in the hypothalamus (*Campbell et al., 2017*), suggesting that some neurons may have hybrid functionalities. These mixed neurons share many transcriptomic markers' expression with many other cell identities and may require the use of other technologies to fully identify them anatomically in the brain.

To perform a cross-species analysis, we generated a rat-based DVC atlas and compared it with the mouse DVC. Our rat dataset was smaller than the mouse data; it is possible some of the new neuronal cell identities (e.g. M_sema, M4) may contain cells belonging to multiple cell types, as treeArches did not append these identities to the tree. Despite this, we corroborated two new cell types in the rat DVC that are not present in mice: the *Pdgfa*-expressing (ortus-akin class) and the immunity-related neurons. *Pdgfra* is a well-described marker gene for OPCs and fibroblasts within the central nervous system (*Kang et al., 2010*) but, to our knowledge, has not been associated with neurons. Notably, although the PDGFRA immuno-signal in HuC/D+ neurons is clearly discernible, it is less intense than in OPCs and fibroblasts, in agreement with expression levels from our snRNA-seq data. Since it was previously demonstrated that OPCs can give rise to neurons in the adult brain (*Robins et al., 2013*) and there is evidence of neurogenesis in the AP following amylin administration in rats (*Liberini et al., 2016a*), it is possible that these PDGFRA+/HuCD+ cells in the AP represent a transient stage of PDGFRA+ OPCs differentiating into neurons.

Regardless of their maturity, we speculate that the Pdgfra neurons (ortus-akin class) have roles in appetite regulation as they express leptin receptor and *Gfral*. Interestingly, while multiple studies have failed to detect *Lepr* via Cre-based reporters (*Cheng et al., 2020b*; *Wada et al., 2014*) or

leptin-induced phosphorylated STAT3 in the AP of mice (*Münzberg et al., 2004*), *Lepr* and leptin-induced phosphorylated STAT3 are detectable within the rat AP (*Smith et al., 2016*; *Liberini et al., 2016b*; *Huo et al., 2007*). We believe one explanation for this species discrepancy in *Lepr* expression is the absence of the ortus class neurons in mice. Future studies are needed to characterize these cells and determine whether their presence is found in other mammals including humans.

DVC-targeting obesity therapeutics, including amylin and GLP-1 mimetics, produce clinically relevant weight loss but are associated with multiple gastrointestinal adverse events and there is a need to develop new obesity therapies that can reduce appetite without unwanted gastrointestinal side effects (*Wilding et al., 2021*). Indeed, the DVC cells that control appetite suppression and nausea are separable (*Huang et al., 2024*; *Cheng et al., 2020a*), making targeting such cells an attractive target for pharmacotherapy. As meal consumption is generally not associated with gastrointestinal distress, we sought to identify refeeding responsive cells within the mouse DVC. From this analysis, we found nearly all neural cell identities are transcriptionally altered by meal consumption. We also found the magnitude of transcriptional changes was relatively small, regardless of the cell class examined. While relatively small, the scale of the transcriptional changes in response to refeeding is similar to many of the gene expression changes observed in *Agrp*-neurons following 16 hr food deprivation (*Steuernagel et al., 2022*). Regardless of whether this is a technical limitation of snRNA-seq methods or accurately reflects the scale of transcriptional changes in the DVC, utilizing transcriptional responses to refeeding to identify neural classes as targets for pharmacotherapy is unlikely to yield specific and targetable pathways.

In general, one limitation in our study is that it is based on nuclear RNA which may reflect a fraction of biological changes. Using nuclear isolation facilitates generating to have single-neuron data (*Lake et al., 2016*) since obtaining viable whole neurons (including all their processes) is very challenging. Previous studies on the relationship between nuclear and cellular RNA content in brain show that nuclear sequencing reads may map to intronic regions of genes in higher proportion (*Lake et al., 2017*). Although this does not appear to impact gene expression analysis for neurons in the current pipelines like the ones we used for our mouse data (*Lake et al., 2017*), it is possible that some biological features such as microglial activation state are not fully recovered using nuclear RNA (*Thrupp et al., 2020*). Although we performed subclustering in the mouse microglia and assessment of the resulting marker genes for each group, we did not find evidence of activation possibly due to this matter. Separately, for meal consumption, we were unable to analyze responses in individual mice as pooling DVCs from individual mice was the most amenable option to maximize nuclei for in our pipeline. As we pooled DVCs from both sexes, this limited our ability to detect sexual dimorphic cell class proportions and gene expression levels.

As the amount of snRNA-seq data performed on the DVC continues to grow, there is a need for a common, scalable label set that can be applied to new studies and integrate new sequencing data. To develop such a label set, we started with a de novo DVC atlas, we generated three layers of cellular granularity for both neural and non-neuronal cell types. Then, using the treeArches pipeline (*Michielsen et al., 2023*), we harmonized our snRNA-seq labels with the Ludwig dataset (*Ludwig et al., 2021*) and increased our cellular granularity from three to four layers of resolution. Manually curating and harmonizing our dataset enhanced the ability of our cell atlas to capture the complexity of the DVC neurons since not all groups seem to contain easily delineable subgroups of cells. For example, 11 of the labels incorporated into our cell hierarchy in this pipeline belong to the monoaminergic neuronal class, which seems to encompass multiple subgroups of specialized cells. We further constructed the rodent DVC cell atlas, which resulted in a fifth layer of cell identity to the monoaminergic M3 neurons. Although the number of identities increases as layers are more specific, this gives this unified atlas greater adaptability to different research needs. Additionally, our labels can be accurately harmonized with new databases regardless of size, and new DVC snRNA-seq data can be added to the tree to generate new branches. Thus, our unified rodent DVC atlas is highly amenable to future research elucidating new biologic and therapeutic insights into DVC cell types.

## Materials and methods

### Experimental design

To generate a snRNA-seq based atlas of the mouse dorsal vagal complex (DVC), we isolated the DVC from 30 adult C57BL/6J mice and subjected them to 10x Genomics-based single-nuclei RNA barcoding and sequencing. Mice were either fasted overnight (n = 5), fasted overnight and then refed 2 hr prior to euthanasia (n = 10), fed ad libitum (n = 5), injected with LiCl (n = 5), or injected with vehicle (n = 5). Conserved cell clusters across these groups were then identified.

### Animals

All animals were bred in the Unit for Laboratory Animal Medicine at the University of Michigan and the Research Institute of the McGill University Health Centre under protocol number MUHC-8223. Procedures performed were approved by the University of Michigan Committee on the Use and Care of Animals, in accordance with Association for the Assessment and Approval of Laboratory Animal Care and National Institutes of Health guidelines. Alternatively, procedures were approved by the animal care committee of McGill University.

### Mouse tissue for snRNA-seq assay

Wild-type C57BL/6J mice (Jackson laboratories) were given ad libitum access to food (Purina Lab Diet 5001) and water in temperature-controlled (22°C) rooms on a 12-hr light–dark cycle with daily health checks. Mice were euthanized by isoflurane anesthesia and decapitated, then the brain was removed from the skull and aligned in a brain matrix. The cerebellar cortex was removed and the DVC was micro-dissected and flash frozen in liquid nitrogen (*Figure 1—figure supplement 4*). To limit any circadian-driven gene expression changes, all animals were euthanized 4 hr following the onset of the light phase. Frozen tissue was stored at –80°C until use.

For assessment of feeding-related gene expression, mice were either ad libitum fed or food was removed at the onset of the dark cycle and the animals were fasted overnight. Animals were then euthanized the next morning under ad libitum fed or fasting conditions. A subset of fasted animals (n = 5) was provided food and allowed to refeed for 2 hr prior to euthanasia.

### Mouse and rat tissue collection for in situ hybridization and immunofluorescent assays

Animals were provided with ad libitum access to food (Purina Lab Diet 5001) and water in temperature-controlled (22°C) rooms on a 12-hr light–dark cycle with daily health status checks. C57BL/6J mice (Jackson laboratories) or Sprague Dawley rats (Jackson laboratories) were euthanized by isoflurane anesthesia followed by $CO_2$ asphyxiation. Animals were then trans-cardially perfused with phosphate-buffered saline for 3 min followed by 5 min perfusion with 10% formalin. Brain tissue was then collected and postfixed for 24 hr in 10% formalin before transfer into 30% sucrose for a minimum of 24 hr. Brains were then sectioned as 30 µm thick, free-floating sections.

### Rat tissue collection for snRNA-seq assays

Wild-type Sprague Dawley rats (Charles River) were provided with ad libitum access to food (Purina Lab Diet 5001) and water in temperature-controlled (22°C) rooms on a 12-hr light–dark cycle with daily health checks. Rats were euthanized by intraperitoneal injections of pentobarbital and decapitated, then the brain was removed from the skull and aligned in a brain matrix. The cerebellar cortex was removed and the DVC was dissected and flash frozen in liquid nitrogen. Frozen tissue was stored at –80°C until use.

### Isolation of nuclei from DVC tissue

Frozen tissue was pooled (five pooled DVCs from mice and two DVCs from rats) and homogenized in Lysis Buffer (EZ Prep Nuclei Kit, Sigma-Aldrich) with Protector RNAase Inhibitor (Sigma-Aldrich) and filtered through a 30-mm MACS strainer (Miltenyi). Filtered samples were centrifuged at 500 rcf for 5 min at 4°C, and pelleted nuclei were resuspended in fresh-made wash buffer (10 mM Tris Buffer, pH 8.0, 5 mM KCl, 12.5 mM $MgCl_2$, 1% BSA with RNAse inhibitor) before undergoing a second filtration and centrifugation. The pelleted nuclei were resuspended in wash buffer with propidium

iodide (Sigma-Aldrich) and underwent fluorescence-activated cell sorting (FACS) on a MoFlo Astrios Cell Sorter to remove debris. PI+ nuclei were collected and centrifuged at 100 rcf for 5 min at 4°C and resuspended in wash buffer to obtain a concentration of 750–1200 nuclei/µL in preparation for sequencing.

## Single-nuclei RNA-sequencing

Library preparation was performed by the Advanced Genomics Core at the University of Michigan. RT mix was added to target approximately 10,000 nuclei recovered per sample and loaded onto the 10×Chromium Controller chip. The Chromium Single Cell 3′ Library and Gel Bead Kit v3, Chromium Chip B Single Cell kit, and Chromium i7 Multiplex Kit were used for subsequent RT, cDNA amplification, and library preparation, as instructed by the manufacturer. Libraries were sequenced on an Illumina NovaSeq 6000 (pair ended with read lengths of 150 nt).

## Processing of the snRNA-sequencing files

From the snRNA-seq, we obtained four FASTQ files (two per sequencing lane, containing forward and reverse sequences, respectively) per mouse treatment and two files with the forward and reverse sequences for the rat samples. We pre-processed the murine FASTQ files using either CellRanger 5.0.1 with inclusion of introns or CellRanger 7.0.1 with default parameters (which include intronic sequences) to align sequencing reads to the murine genome (*Zheng et al., 2017*). We aligned to the *Mus musculus* refdata-gex-mm10-2020-A reference genome. For the rat files, we used CellRanger 3.0.1 with no intron inclusion and aligned using the *Rattus norvegicus* genome assembly Rnor_6.0 (*Zheng et al., 2017*). We obtained 110,167 and 13,360 cells from the murine and rat experiments, respectively. In addition to these two datasets, we also used the following steps to process the murine Dowsett dataset (*Dowsett et al., 2021*) publicly available as a raw gene expression matrix set of files. We processed the gene expression matrices in R (*R Development Core Team, 2023*) with Seurat v5 and SeuratObject v5 to remove low-quality cells (*Hao et al., 2024*). For processing and analysis, R v4.3.1 (*R Development Core Team, 2023*) was used. We filtered cells with ≥500 number of genes mapped and >1.22 RNA unique molecular identifiers (UMIs) counts/genes mapped ratio (based on the first 2 percentiles per sample) (*Supplementary file 10*). Subsequently, we merged the data per treatment, calculated the median RNA content of clusters at resolution 1.0, and those below the first quartile of the median distribution were removed (*Supplementary file 11*). Additionally, we used DoubletFinder v2.0.3 based on the expected doublet rates by 10X Genomics after adjusting for homotypic doublets modeled as the sum of squared annotation frequencies (*McGinnis et al., 2019*; *Supplementary file 12*). On each iteration, we processed the data with libraries scaled to 10,000 UMIs per cell and log-normalized. We identified the most variable genes by computing a bin $Z$-score for dispersion based on 20 bins average expression. We regressed UMI counts and used principal component (PC) analysis for dimensionality reduction on to the top 2000 most variable genes. We used the first 30 PCs for $k$-nearest neighbors clustering and for uniform manifold approximation and projection (UMAP) projections using Seurat v5 (*Hao et al., 2024*) default parameters. Murine samples integration in a single matrix was done with Harmony (*Korsunsky et al., 2019*), after which clustering and UMAP projections were performed using the harmony embeddings instead of PCs.

## Mapping of our murine data to existing brain databases

We mapped the resulting murine integrated high-quality singlets to the celldex v1.12.0 built-in mouse RNA sequencing reference and other four mouse databases (*Campbell et al., 2017*; *Chen et al., 2017*; *Romanov et al., 2017*; *Tasic et al., 2016*; *Supplementary file 1*) using SingleR v2.4.1 (*Aran et al., 2019*) and scRNA-seq v2.16.0 (*Risso and Cole, 2025*) packages in R (*R Development Core Team, 2023*). Since each database has its own cell type nomenclature, in our murine dataset, we established a 'likely-cell type' to be unanimous among all databases, for example, if all databases labeled a cell as 'neuron'. We initially labeled the cells with conflicting or unassigned label by one or more databases as 'unknown'. To confirm the cell type of all the cells labeled and to identify the unknown cell identities, we mapped 473 markers from the literature. For this, the average expression per marker gene was obtained on scaled data for each cluster at resolution 1.0 (i.e. 48 clusters) (*Supplementary file 2*). We also manually visualized UMAP projections from all these markers in our dataset. Some clusters (e.g. cluster 27) showed in the previous step to contain cells belonging to different cell types, so we subset

and reprocessed these clusters using the Harmony embeddings to assign their cell identity based on marker expression at the lowest resolution level that allowed us to separate them.

## Assigning high-resolution clustering-based cell identity in the murine dataset

In the murine dataset, for all non-neuronal clusters at resolution 1.0 gene expression markers were obtained using the FindConservedMarkers function from Seurat v5 (*Hao et al., 2024*) package in R (*R Development Core Team, 2023*). The function was run using the MAST algorithm (*Finak et al., 2015*) on scaled data with a log FC threshold of 0.25. Only upregulated marker genes detected on 40% of all cells in that cluster were retained. Since the FindConservedMarkers gives a $\log_2$FC output per sample, we calculated the average $\log_2$FC across samples and the proportion of samples for which a gene had an adjusted p-value ≤0.05. A marker gene was considered as such if it had an average $\log_2$FC >1 across samples and was significant (i.e. had an adjusted p-value ≤0.05) in more than 80% of the samples. Furthermore, neurons were subset, reprocessed, and subjected to new clustering through Seurat v5 (*Hao et al., 2024*). We obtained the gene markers per cluster at resolution 1.0 as described. We named the clusters based on the known functions of the upregulated/downregulated genes (e.g. myelinating oligodendrocytes), peculiarities of the cell groups (e.g. $Ca^+$-permeable astrocytes) or their upregulated gene expression markers if the previous methods were unsuitable (e.g. Ano2 neurons) (*Supplementary file 6*). All visualizations, unless otherwise stated, were done using ggplot2 v3.5.0 and ComplexHeatmap v2.18.0 packages in R (*R Development Core Team, 2023*).

## Oligodendrocytes trajectory inference

We subset the oligodendrocytes (i.e. pre-myelinating, myelinating intermediate, and myelinating) in our murine dataset and placed each cell on an inferred cellular trajectory using their transcriptomic data in R (*R Development Core Team, 2023*). We used the SCORPIUS v1.0.9 package (*Cannoodt et al., 2016*) with default parameters to perform dimensionality reduction by PCA on log-normalized counts and to obtain the trajectory image, with a random seed of 1000. We additionally performed the analysis on the Harmony embeddings from the original data integration.

## Gene expression files format interconversions

To convert Seurat (RDS) DVC datasets to 'h5ad' format and vice versa, we used reticulate v1.35.0 (*Ushey et al., 2024*) and anndata 0.7.5.6 (*Cannoodt, 2023*) packages in R (*R Development Core Team, 2023*) with import of scipy, scanpy, numpy, and anndata modules from Python v3.9 (*Van Rossum and Drake, 2009*). To convert .txt, .csv, and .tsv datasets (i.e. the Ludwig dataset [*Ludwig et al., 2021*], the spinal cord dataset [*Sathyamurthy et al., 2018*], and the cortex/hippocampus data-base [*Batiuk et al., 2020*], we converted those to 'h5ad' format in Python v3.9 [*Van Rossum and Drake, 2009*]).

## Murine DVC hierarchy construction

We converted our murine labeled count matrix to h5ad. We used treeArches (*Michielsen et al., 2023*) in Python v3.9 (*Van Rossum and Drake, 2009*) to create a manual tree using the three layers of cellular granularity in our database. We reprocessed the samples using the treeArches pipeline (*Michielsen et al., 2023*) to normalize the count data (counts normalized per cell and then log-normalized). After multiple tests, we determined that the 2500 most variable genes were the optimum parameter for the integration of other datasets based on our murine data; therefore, we identified the top 2500 most variable genes and integrated the samples using scVI and treeArches (*Michielsen et al., 2023*; *Gayoso et al., 2022*). This reference latent space obtained after integration was used to generate the UMAP embeddings. This sample integration was done to ensure that inter-sample variations were removed for the cell identity steps. We trained our manual tree based on the cell identity layer-3 labels using default parameters.

We subset the Ludwig dataset (*Ludwig et al., 2021*) to match the initial 2500 most variable genes from our datasets and performed surgery to incorporate labels from the Ludwig dataset (*Ludwig et al., 2021*). The two layers of granularity established by *Ludwig et al., 2021* were combined in the variable 'identity_layer3' to contain the highest granularity of cell identity for neurons and non-neuronal cell types. Next, we normalized the count data as specified for our murine dataset

and mapped the Ludwig dataset on the reference latent space using scArches. We then used the learn_tree function with default parameters for hierarchical progressive learning from scHPL v1.0.5 (*Michielsen et al., 2021*), which yielded a hierarchy of harmonized cell labels from both datasets. We printed all trees using matplotlib v3.8.2 (*Hunter, 2007*) and scHPL v1.0.5 (*Michielsen et al., 2021*). All steps performed and scripts used in this phase are available in the GitHub repository: https://github.com/LabSabatini/DVC_cell_atlas (copy archived at *Hes et al., 2024*).

## Corroboration of the validity of our murine cell hierarchy using treeArches prediction modality

Using treeArches (*Michielsen et al., 2023*) and its dependencies in Python v3.9 (*Van Rossum and Drake, 2009*), we compared the original cell labels in our and Ludwig datasets with the resulting prediction of the label for each cell using our learned hierarchy. We used the predict_labels function from scHPL v1.0.5 (*Michielsen et al., 2021*) with default parameters, and the latent representation obtained when we constructed our murine hierarchy. We compared the original and the predicted labels through a heatmap visualization. Next, we predicted the labels of new data, a murine dataset by Dowsett which comprises the NTS, a part of the DVC (*Dowsett et al., 2021*). We processed the Dowsett data similarly to the Ludwig dataset and obtained the query latent representation. We then used the predict_labels function from scHPL v1.0.5 (*Michielsen et al., 2021*) with default parameters to transfer the labels in our hierarchy to the Dowsett dataset based on gene expression similarity among the 2,500 initial most variable genes. We then compared the resulting labels from this prediction to the manual mapping of our cell identities based on marker expression (*Supplementary file 4*) using UMAP embeddings in R (*R Development Core Team, 2023*).

## Labeling of our rat DVC dataset

We processed our rat dataset as previously described to obtain high-quality singlets. Based on the three layers of cell identity that we created to label the murine data from our experiments, we calculated the average expression of the top 5 marker genes for each of the layer-3 identities (except duplicated genes and mouse-specific genes) on every rat cluster to identify those that were equivalent and could be labeled as the murine data (*Supplementary files 4 and 6*). Since this dataset is smaller than the mouse data, we decided to use clusters at a resolution of 2.0 to capture more heterogeneity and delineate better the non-neuronal cell populations. Clusters 27 and 29 were further subset and reclustered since we found them to contain a mix of endothelial cells, mural cells, tanycytes, and fibroblasts based on marker expression. Microglia and mixed neurons were subclustered as well. The lowest resolution level that allowed for separating populations was used to label the cells. Neurons were also subset, and we mapped the murine neuronal classes markers to resolution 2.0 markers, which were manually curated to delineate better the different populations within them. For all cell identities, and because we only applied one treatment to the rats (i.e. fed ad libitum), we used the FindMarkers function from Seurat v5 (*Hao et al., 2024*) package in R (*R Development Core Team, 2023*) to obtain gene expression markers of each rat cell population. The function was run using the MAST algorithm (*Finak et al., 2015*) with the same parameters used for the murine data (*Supplementary file 3*).

## Construction of the rodent DVC hierarchy

We then integrated our rat labeled dataset to the murine samples in Python v3.9 (*Van Rossum and Drake, 2009*) using the initial top 2500 most variable genes initially defined in our mouse data. Incorporation of rat labels was done as previously described for the Ludwig dataset using the learn_tree function for hierarchical progressive learning from scHPL v1.0.5 (*Michielsen et al., 2021*). The cell populations that were not rejected but placed at the bottom of the tree were moved manually to their belonging parent branches. For example, the 'smooth_muscle_rat' label was moved to be a children node of the mural cells node. If the parent branch did not exist, we created it. For example, the 'immunity_related_rat' label belonged to a new rat neuronal class; therefore, we incorporated this parent node within the neurons node, and then we moved the 'immunity_related_rat' label to be a children node. After manual curation, we re-trained the tree. We printed all trees using matplotlib v3.8.2 (*Hunter, 2007*) and scHPL v1.0.5 (*Michielsen et al., 2021*). All steps performed and scripts used in this phase are available at the GitHub repository https://github.com/LabSabatini/DVC_cell_atlas (copy archived at *Hes et al., 2024*).

## Mapping gene expression in other brain-site snRNA-seq datasets

We downloaded labeled datasets from six brain regions different from the DVC: the HypoMap, spinal cord, cortex, hippocampus, pons, and forebrain databases (*Steuernagel et al., 2022*; *Batiuk et al., 2020*; *Jessa et al., 2019*; *Sathyamurthy et al., 2018*). For the HypoMap data, we subset the database by randomly selecting cells belonging to 32 included hypothalamic samples. We processed the datasets in Seurat v5 (*Hao et al., 2024*) as previously described and obtained Harmony (*Korsunsky et al., 2019*) embeddings to obtain UMAP projections. The labels used in the visualization are the original curated labels provided on each dataset. Neurons were further subset from the processed datasets and reprocessed as previously described. The other datasets were processed in the same manner after subsetting astrocytes. The pons and forebrain data were downloaded using scRNAseq v2.16.0 packages in R through SingleR v2.4.1 (*Aran et al., 2019*).

## Differential gene expression analysis in murine treatments

We subset the murine dataset neurons and glial cells at layer-2 granularity of identity and obtained differential gene expression for pair-wise comparison of treatments (i.e. refed versus ad libitum fed, refed versus fasted, and LiCl versus vehicle injected mice). We used the FindMarkers function from Seurat v5 (*Hao et al., 2024*) package in R (*R Development Core Team, 2023*) using the MAST algorithm (*Finak et al., 2015*) on log-normalized counts with a minimal log FC threshold of 0.1 and default parameters. The number of cells per treatment can be found in *Supplementary file 8*.

## Immunofluorescent staining

Free-floating brain sections from mouse and rat were washed with PBS, three times for 5 min. The sections were then blocked for 1 hr in PBS containing 0.1% Triton X-100 and 3% normal donkey serum (Fisher Scientific). The sections were incubated overnight at room temperature with Goat anti-PDGFRA (Invitrogen, CAT#, 1:200), HuC/HuD (Invitrogen, A-21271, 1:30). The following day, sections were washed and incubated for 2 hr with Alexaflour 488 and Cy3-conjugated secondary antibody (Jackson Immunoresearch, 1:500). Tissues were then washed three times in PBS before being mounted onto glass slides, covered in mounting media (Fluoromount-G, Southern Biotechnology) and coverslipped. Imaging was performed using an Olympus BX61 microscope and a Zeiss LSM780-NLO laser scanning confocal microscope equipped with IR-OPO lasers at the Molecular Imaging Platform, Research Institute of the McGill University Health Centre (RI-MUHC), Montreal, CA.

## In situ hybridization and imaging

Brain sections containing the DVC from four wild-type C57BL/6J mice obtained as described above were fixed on glass slides and stored at –20°C for no more than 36 hr. We followed the manufacturer's ACD 323100 user manual for the RNAscope Multiplex Fluorescent Reagent Kit v2 Assay for fixed frozen tissue samples. Briefly, we rinsed the slides with 1X PBS and incubated them at room temperature for 10 min after adding ~1–2 drops of $H_2O_2$ per section. We removed the $H_2O_2$ from slides and rinsed them twice with distilled water (di$H_2O$). Next, we submerged the slides in 200 ml of hot di$H_2O$ for 10 s followed by 200 ml of RNAscope 1X Target Retrieval Reagent for 5 min, using a steamer with lid. After briefly transferring the slides to room temperature di$H_2O$, we submerged them in 100% ethanol for 3 min and allowed them to air dry. We applied approximately 1–2 drops of RNAscope Protease III to each section and incubated them at 40°C for 30 min in a HybEZ oven with di$H_2O$ wet paper in the tray. After rinsing the slides with di$H_2O$, we hybridized the probes at 40°C for 2 hr.

From this point on, we performed all steps by incubating the slides at 40°C in the HybEZ oven with humid tray followed by rinse using RNAscope 1X wash buffer. We applied approximately 1–2 drops of RNAscope reagents AMP1, AMP2, AMP3, and then intercalated HRP channel (15 min), fluorophore (30 min), and HRP blocker (15 min) for each channel present in the probe mix. The mix of probes used for the *Th/Cck* co-expression assay was RNAscope Mm-Th-C2 (317621-C2) and Mm-Cck-C3 (402271-C3) diluted in probe diluent as specified in the user manual. For *Hcrt*, we used RNAscope Mm-Hcrt-C2 (490461-C2) and for *Agrp*, Mm-Agrp (400711), in a dilution as specified in the user manual. The used fluorophores were Cy3 for *Cck* and *Hcrt* (1:1,500), and Cy5 for *Th* and *Agrp* (1:1,000) diluted in RNAscope TSA buffer. DAPI dye (1:1,000) was applied at the end of the assay, and the samples were stored for ~48 hr at –4°C, protected from light, before imaging. We imaged the sections using a Zeiss

LSM780-NLO laser scanning confocal microscope with IR-OPO lasers at the Molecular Imaging Platform at the RI-MUHC, Montreal, CA.

## Acknowledgements

We would like to thank the authors of the Ludwig and the Dowsett publications for making publicly available their datasets. We thank M Myers Jr. for the valuable help provided; S Hebert and S Bailey for their insights and suggestions in regards to the bioinformatic analysis; members of the Kokoeva lab for their insights on the in situ hybridization and immunohistochemistry pipelines; S Feng and M Fu for technical assistance for images acquisition at the Molecular Imaging Platform at the RI-MUHC, and A Duensing for assistance with relevant file transfer. This research was also supported by grants from Canadian Institutes for Health Research (PJT180590 for PVS and 202010PJT for MK) and the Natural Sciences and Engineering Research Council of Canada (RGPIN-2022-03390). HJM was funded in part by the Canada First Research Excellence Fund and Fonds de recherche du Quebec, awarded to the Healthy Brains, Healthy Lives initiative.

## Additional information

### Funding

| Funder | Grant reference number | Author |
| --- | --- | --- |
| Natural Sciences and Engineering Research Council of Canada | RGPIN-2022-03390 | Paul V Sabatini |
| Canadian Institutes of Health Research | 202010PJT | Maia V Kokoeva |
| Canadian Institutes of Health Research | PJT180590 | Paul V Sabatini |
| Canada First Research Excellence Fund | Healthy brains, healthy lives | Hunter J Murdoch |
| Fonds de recherche du Québec | Healthy brains, healthy lives | Hunter J Murdoch |

The funders had no role in study design, data collection, and interpretation, or the decision to submit the work for publication.

### Author contributions

Cecilia Hes, Conceptualization, Data curation, Formal analysis, Investigation, Writing - original draft, Writing - review and editing; Abigail J Tomlinson, Fatemeh Soltani, Data curation, Writing - review and editing; Lieke Michielsen, Formal analysis, Methodology, Writing - review and editing; Hunter J Murdoch, Formal analysis, Writing - review and editing; Maia V Kokoeva, Resources, Writing - review and editing; Paul V Sabatini, Conceptualization, Data curation, Funding acquisition, Methodology, Writing - original draft, Writing - review and editing

### Author ORCIDs

Cecilia Hes ⓘ https://orcid.org/0000-0002-4828-3260
Paul V Sabatini ⓘ https://orcid.org/0000-0001-6613-566X

### Ethics

All animals were bred in the Unit for Laboratory Animal Medicine at the University of Michigan and the Research Institute of the McGill University Health Centre under protocol number MUHC-8223. Procedures performed were approved by the University of Michigan Committee on the Use and Care of Animals, in accordance with Association for the Assessment and Approval of Laboratory Animal Care and National Institutes of Health guidelines. Alternatively, procedures were approved by the animal care committee of McGill University.

Reviewer #1 (Public review): https://doi.org/10.7554/eLife.106217.3.sa1
Reviewer #2 (Public review): https://doi.org/10.7554/eLife.106217.3.sa2
Author response https://doi.org/10.7554/eLife.106217.3.sa3

## Additional files

### Supplementary files

Supplementary file 1. Murine brain databases used to map our unlabeled mouse dataset. We used four databases included in the scRNAseq v2.16.0 (*Marques et al., 2016*; *Campbell et al., 2017*; *Chen et al., 2017*; *Romanov et al., 2017*; *Risso and Cole, 2025*) R package and the celldex v1.12.0 built-in mouse RNA-seq reference (which we called 'SingleR' database) using SingleR (*Aran et al., 2019*) function to transfer labels to our dataset in R (*R Development Core Team, 2023*). The name, brain area, and reference in which each database was made public are specified.

Supplementary file 2. Literature-derived markers per central nervous system cell identity. We thoroughly searched in the literature for established markers at the mRNA and/or protein level. Some markers were found to be described for more than one cell population which are specified in columns 'notes' and 'notes_2'. After initial labeling of our murine dataset with the databases as described, we proceeded to obtain the average expression (on log-normalized counts) of each gene coding for the markers for the labeled cells (i.e. astrocyte, endothelial, microglia, neuron, and oligodendrocyte) and unlabeled cells (i.e. unknown). We also calculated the average expression for each cluster in our dataset, established during our processing pipeline, at resolution 1.0.

Supplementary file 3. Description of our DVC layer-3 neuronal cell identities established in rats. Rat neurons were subset and reprocessed before labeling at layers 2 and 3 of granularity. We used neuronal class gene expression markers to identify clusters at resolution 2.0 that belonged to each neuronal cell class. These cell classes were further subclustered, and markers from *Supplementary file 6* and UMAP projections were used to label each population in the dataset.

Supplementary file 4. Literature-derived markers per central nervous system cell identity. Breakdown of the cell identities (except neurons at layer-3 of cellular resolution) we established and corroborated using a combination of markers surveyed in *Supplementary file 11* and additional markers established using the FindConservedMarkers function from Seurat v5 (*Hao et al., 2024*) package, indicating a cell state derived from the literature. For example, 's_fibroblasts' were named after finding high expression of *Stk39*, a stress-related marker (*Kasai et al., 2022*).

Supplementary file 5. Expression of neurotransmitter-related genes in our murine neurons. Expression of genes coding for neurotransmitter synthesis enzymes TH, DBH, TPH, GABA, NOS, prohormone SST, and primary neurotransmitter transporters VMAT2, VGLUT1, VGLUT2, GLYT2 per neuron in our dataset. Expression and glutamate and GABA release are specified as binary variables (i.e. 0 = no expression; 1 = expression) in the green and blue columns, respectively. The percentile of the log-normalized expression is shown. ~55% of the neurons display expression of genes related to the synthesis/release of >1 primary neurotransmitter.

Supplementary file 6. Description of our DVC layer-3 neuronal cell identities established in mice. Neurons were subset and reprocessed before labeling at layer-3 granularity. The MAST algorithm (*Finak et al., 2015*) was used through Seurat v5 (*Hao et al., 2024*) on scaled data with a log fold-change threshold of 0.25 and detection threshold on ≥40% of cells on each neuronal cluster at resolution 1.0. We searched the resulting gene expression markers in the literature and compiled them. We then based the names for our cell identities on these genes.

Supplementary file 7. Expression of the murine DVC identity markers by rat DVC clusters. Average log-normalized expression of the top 5 gene expression markers of each of our murine layer-3 cell identities (except duplicated genes $n = 37$, and mouse-specific genes $n = 27$; total genes included = 186) by rat DVC cluster at resolution 2.0.

Supplementary file 8. Labeled cells in our murine dataset. Number of cells per treatment per identity layer-2.

Supplementary file 9. Receptor expression in magnaclasses 1 and 2 between meal-related treatments in mice. Differential expression analysis (MAST algorithm) (*Finak et al., 2015*) results for genes coding for metabolism-related receptors between meal-related treatments (i.e. refed versus ad libitum fed, and refed versus overnight fasted) and per neuronal magnaclass. If a receptor is missing for any given comparison for any of the magnaclasses, the log fold-change of that

gene between both treatments in that neuronal class was <0.1; therefore, it is not reported (see **Methods**).

Supplementary file 10. Filtering of our and Dowsett datasets during processing. Compilation of statistics and cells processed for the initial high-quality cell filtering per sample.

Supplementary file 11. Cluster filtering of our and Dowsett datasets during processing. Compilation of low-quality cluster analysis statistics to filter high-quality clusters per sample.

Supplementary file 12. Singlets retained per dataset after processing. Number of doublets and singlets identified by DoubletFinder (*McGinnis et al., 2019*) after initial filtering of our and Dowset dataset per sample. Doublets were removed from the datasets before sample integration with Harmony (*Korsunsky et al., 2019*) (this step was not performed in the rat dataset as it was only one sample, *n* = 2 rat DVC).

MDAR checklist

## Data availability

All data needed to evaluate the conclusions in the paper are present in the paper and/or the Supplementary Materials. FASTQ files have been deposited in the NCBI Sequence Read Archive (Accession: PRJNA1161292). In addition, our murine and rat datasets have been made available for visualization and analysis through the Broad Institute Single Cell Portal (https://singlecell.broadinstitute.org/single_cell/study/SCP2773). The labels from our cell atlas can be transferred to new datasets through treeArches using the available hosted Jupyter Notebook in Google Colab for the mouse data (https://colab.research.google.com/drive/1v983gcQxIwBN-4lGx18JXPweZoKqMbwX#scrollTo=SJTZppZWa63), and for the rodent data (https://colab.research.google.com/drive/10tAmfRosMscuBak6wHX80TgdEBsHeWQB#scrollTo=941ab5c1-8910-4e67-a632-4d18601d45b4). Original code used for constructing the murine and rodent cell hierarchies and predicting labels using treeArches is available in the GitHub repository https://github.com/LabSabatini/DVC_cell_atlas (copy archived at *Hes et al., 2024*).

The following dataset was generated:

| Author(s) | Year | Dataset title | Dataset URL | Database and Identifier |
|---|---|---|---|---|
| Hes C, Tomlinson AJ, Michielsen L, Murdoch HJ, Soltani F, Kokoeva M, Sabatini PV | 2025 | A unified rodent atlas reveals the cellular complexity and evolutionary divergence of the dorsal vagal complex | https://www.ncbi.nlm.nih.gov/bioproject/?term=PRJNA1161292 | NCBI BioProject, PRJNA1161292 |

The following previously published datasets were used:

| Author(s) | Year | Dataset title | Dataset URL | Database and Identifier |
|---|---|---|---|---|
| Ludwig MQ, Cheng W, Gordian D, Lee J | 2021 | A genetic map of the mouse dorsal vagal complex and its role in obesity | https://www.ncbi.nlm.nih.gov/geo/query/acc.cgi?acc=GSE166649 | NCBI Gene Expression Omnibus, GSE166649 |
| Dowsett GKC, Tadross JA, Lam BY, Cimino I | 2021 | A survey of the mouse hindbrain in the fed and fasted states using single-nucleus RNA sequencing | https://www.ncbi.nlm.nih.gov/geo/query/acc.cgi?acc=GSE168737 | NCBI Gene Expression Omnibus, GSE168737 |

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
