## [Editor Report · eLife Assessment]

This manuscript applies state-of-the-art techniques to define the cellular composition of the dorsal vagal complex in two rodent species (mice and rats). The result is a **fundamental** resource that substantially advances our understanding of the dorsal vagal complex's role in the regulation of feeding and metabolism while also highlighting key differences between species. The analyses of single-cell profiling experiments in the manuscript provide **compelling** insight into the cellular architecture of the dorsal vagal complex, with potential implications for obesity therapeutics. [Editors’ note: The FASTQ files of the rat snRNA-Seq data for this manuscript are not available as the authors could not locate the files after moving institutions. However, the count matrices are available to download via the Broad Single cell portal.]

---

## [Referee Report · Reviewer #1 (Public review)]

Summary:

This paper is using state-of-the-art techniques to define the cellular composition and its complexity in two rodent species (mice and rats). The study is built on available datasets but extends those in a way that future research will be facilitated. The study will be of high impact for the study of metabolic control.

Strengths:

After revision, the paper is much improved. I have no further comments.

---

## [Referee Report · Reviewer #2 (Public review)]

In this manuscript, Hes et al. present a comprehensive multi-species atlas of the dorsal vagal complex (DVC) using single-nucleus RNA sequencing, identifying over 180,000 cells and 123 cell types across five levels of granularity in mice and rats. Intriguingly, the analysis uncovered previously uncharacterized cell populations, including Kcnj3-expressing astrocytes, neurons co-expressing Th and Cck, and a population of leptin receptor-expressing neurons in the rat area postrema, which also express the progenitor marker Pdgfra. These findings suggest species-specific differences in appetite regulation. This study provides a valuable resource for investigating the intricate cellular landscape of the DVC and its role in metabolic control, with potential implications for refining obesity treatments targeting this hindbrain region.

In line with previous work published by the PI, the topic is of clear scientific relevance, and the data presented in this manuscript are both novel and compelling. Additionally, the manuscript is well-structured, and the conclusions are robust and supported by the data. Overall, this study significantly enhances our understanding of the DVC and sheds light on key differences between rats and mice.

I have reviewed the revised manuscript and am pleased to confirm that the authors have addressed my previous comments and concerns.

---

## [Author Response]

The following is the authors’ response to the original reviews

We thank the expert reviewers for their careful consideration of our manuscript and the feedback to help us strengthen our work. Please find a response to each reviewer’s comments below. We have included the original text from the reviewer in unbolded text and our response, immediately below, in bold text for clarity.

**Reviewer #1:**
(1) Appetite is controlled, not regulated; please reword throughout.

The reviewer raises a valid point that we have misused the word “regulate” in certain instances and “control” would be more accurate term. We have made adjustments throughout the manuscript.

(2) One minor point that would further strengthen the data is a more distinct analysis of receptors that are characteristic of the different populations of neuronal and non-neuronal cells; this part could be improved.

We thank the reviewer for this suggestion as we had not directly compared metabolicallyrelevant peptides/receptors between the mouse and rat DVC. We have included a list of selected receptors and neuropeptides expression (see Figure S13) for neuronal cells in mouse and rat. We have included this figure as a new supplement. There are some interesting insights from this data, including the relatively broad expression of Lepr in the rat compared with the mouse and the absence of proglucagon expressing neurons within the rat DVC.

**Reviewer #2:**
(1) In some of the graphs, the label AP/NTS is used, but DVC would be more appropriate.

We have reviewed the figures and legends to ensure appropriate use of DVC. We thank the reviewer for bringing this oversight to our attention.

(2) Line 124, p7 - Sprague Dawley RATS

We have changed the text to “Sprague Dawley rats”

(3) Line 132, p7 - The phrase "were provided with given access to food" needs grammatical correction.

We agree the text was poorly written. The sentence has been corrected to: “Wild-type Sprague

Dawley rats (Charles River) were provided with ad libitum access to food Purina Lab Diet

1. and water in temperature-controlled (22°C) rooms on a 12-hour light-dark cycle with daily health checks.” We have also reviewed the entire manuscript and made additional amendments where necessary.

(4) Page 15 - Mention that GFAP is a marker for astrocytes. Additionally, correct the typo "gfrap".

We have corrected the misspelling of “Gfap” within the text. We appreciate the reviewer’s comment that there is value in communicating to the nonexpert reader that GFAP is a marker for astrocytes, however, as our data and that from other snRNA-Seq studies show that Gfap mRNA only labels a subset of astrocytes, our preference is to refrain from stating this. Our data suggests the sole use of Gfap as an astrocyte marker will not reflect the true astrocyte population.

(5) Line 432, p15 - What was the rationale for selecting clusters 23, 26, and 27?

We chose to perform subclustering on these clusters because they displayed multiple cell identities when surveyed for the 473 marker genes as described in Methods 2.6. In order to separate these, the granularity was increased in them by sub-clustering.

(6) Line 533, p18 - only 5 out of 34 neurons express GFRAL, which makes the language used a little bit misleading. As per the comment above, I would specify that only a subset (X%) of neurons express GFRAL, and apply the same approach for other markers.

We thank the reviewer for raising this point. We agree the text, as written, was an oversimplification. We adjusted the text as recommended: that a subset (~15%) express detectable Gfral mRNA but is likely an underrepresentation due to the challenges in detecting lowly expressed transcripts such as *Gfral*.

(7) Line 547, p18 - This statement appears to refer to rat data specifically, rather than rodent data in general.

The text has been corrected.

(8) Section 3.6 - The discussion on meal-related transcriptional programs in the murine DVC does not mention Figure S10A and B.

We thank the reviewer for the observation. It is true that we do not discuss this figure. Fig10S is the integration of samples in treeArches, a necessary step to build the hierarchy in python so the learning algorithm uses only genes that are related to identity and not treatment, we obtained the same overlap of samples when we used R to assign identities. This figure demonstrates our integration was successful because it is only considering genes that are not-treatment related to establish identities, those which are expressed by cells regardless of their response to any treatment. For the meal-related analysis, we were interested in the genes that are changed by treatment, and this is why the analysis differed. We have included a sentence in the methods to clarify this point that states: " This sample integration was done to ensure that inter-sample variations were removed for the cell identity steps."

(9) Page 5, citation 10 - the author cited a clinical trial for glucagon and GLP-1 receptor dual agonist survodutide for "DVC neurons' role in appetite and energy balance stems from their role as therapeutic targets for obesity". A more appropriate citation (such as a review) would be preferable.

We appreciate the suggestion by the reviewer. We have updated our references to reflect a recent manuscript from the Alhadeff group which demonstrates the DVC acts as the target of GLP1-based therapies. We have also included a review as suggested 10.1038/s42255-02200606-9.

(10) Line 52, p5 - a citation of obesity is needed, as the current ref only pertains to cancer cachexia.

We have included a reference for obesity.

(11) In the discussion, it would be valuable to elaborate on the potential significance of DVCspecific glial cells (perhaps at the end of the second paragraph?).

We thank the reviewer for this suggestion. Our discovery of a DVC-specific astrocyte transcriptional profile was underrepresented within the discussion. We have attempted to expand this discussion on the suspected roles for these DVC-specific astrocytes. Much of this discussion is based on the distinct localization pattern of Gfap mRNA in the DVC (see Image on Allen Brain ISH) which shows dense signal at the boundary of the AP and NTS. As astrocytes have well established roles in maintaining BBB integrity, it is our speculation that this is a major role of these cells. However, functional studies will be critical to assess the roles of these astrocytes in DVC biology.

(12) Line 683, p22 - Consider adding PMID: 38987598 which describes the dissociable GLP-1R circuits.

We appreciate this recommendation – we have included this reference.

(13) The authors suggest that a possible explanation for the discrepancy between snRNA-Seq and in situ hybridization data is that Agrp and Hcrt mRNA reads in snRNA-Seq overwhelmingly mapped to non-coding regions. To what extent could this limitation affect other genes included in the current analyzed 10x datasets?

As shown by Pool and cols. (https://doi.org/10.1038/s41592-023-02003-w) including intronic reads improves sensitivity and more accurately reflects endogenous gene expression. Therefore, including intronic reads is considered more of a strength than a limitation and is now default in platforms such as CellRanger. While including intronic reads for mapping snRNA-Seq data, we would advise corroboration of snRNA-Seq findings with published literature or detection of coding mRNA or protein. In our case, the detection of hypothalamic neuropeptide via snRNA-Seq data could not be verified by performing in situ hybridizations using probes that detect exons. Therefore, *Hcrt* and *Agrp* having only intronic reads suggest a regulatory (reviewed in https://doi.org/10.3389/fgene.2018.00672) rather than a coding role in the DVC.

(14) Given the manuscript's focus on feeding and metabolism, I believe a more detailed description and comparison of the transcription profile of known receptors, neurotransmitters, and neuropeptides involved in food intake and energy homeostasis between mice and rats would add value. Adding a curated list of key genes related to feeding regulation would be particularly informative.

A similar request was made by reviewer #1. Please see the full response above. Briefly, we have performed additional analysis of the mouse and rat DVC data and included this data as an additional supplemental figure (Figure S13).

(15) Line 479-482, p17 - It would be helpful if the authors could quantify (e.g., number and/or percentage) the extent of TH and CCK co-expression.

We have amended the text of the manuscript to include quantification of Cck and Th colocalization. According to our snRNA-seq data, out of the 764 *Th*-expressing neurons, 80 coexpress *Cck* in the mouse (~10%). The *Cck*-expressing cells are more numerous, 3,821 in total.

(16) The number of animals used differs significantly between species, which the authors acknowledge as a limitation in the discussion. Since the authors took advantage of previously published mouse data sets (Ludwig and Dowsett data sets), I wonder if the authors could compare/integrate any rat data set currently available in rats as well to partially address the sample size disparity.

We agree with the review that our rat database is considerably smaller than our mouse database, making comparisons between rat and mouse DVC challenging. We attempted to increase the size of our rat DVC atlas by incorporating publicly available rat DVC snRNA-Seq data (Reiner et al 2022). However, we found several issues with the quality of this data including low UMIs/cell and gene #/cell. For these reasons, we decided against merging these two datasets. So while relatively small, our rat DVC atlas uses high quality data and serves as a valuable starting point. By introducing TreeArches as a method to relatively easily incorporate new snRNA-Seq data into our own, it is our hope that future studies will do so and thus expand the rat DVC atlas we have built.

(17) In the Materials and Methods section, LiCl is mentioned as one of the treatment conditions; however, very little corresponding data are presented or discussed. Please include these results and elaborate on the rationale for selecting LiCl over other anorectic compounds.

The reviewer is correct, some of the tissues used in this study were from animals treated with LiCl prior to euthanasia. Our intent was to contrast the transcriptional effects induced by LiCl (an anorectic agent with aversive properties) with refeeding (a naturally rewarding and satiating stimuli). However, upon analyzing the data, we found very few transcriptional changes induced by LiCl. It is unclear to us whether this was a technical failure in the experiment and so did not elaborate on the results.

**Reviewer #3 (Recommendations for the authors):**
(1) The use of both sexes is indicated in the discussion, but methods and results do not address sex distribution in the investigated groups. Also, the groups could be more clearly described, e.g., the size of the 2 hour refeeding mouse group varies from n=10 to n=5.

We have clarified the text, in line with the reviewer’s suggestion. There were two cohorts of fasted/ refed mice (n=5 each), so in the manuscript methods it is stated as n=10 because of this. The fasted-only group, which was not refed before euthanasia is a separate group, n=5.

(2) Page 20, the last sentence needs to be reworded.

We thank the reviewer for this recommendation. The text has been amended to improve clarity of the sentence.

(3) Page 22, lines 691-692 - this sentence needs to be reworded.

We thank the reviewer for this comment. The offending sentences have been amended.

(4) While the authors find transcriptional changes in all neuronal and non-neuronal cell types, which is interesting, the verification of known transcriptional changes (e.g., cFos) is unaddressed. cFos is a common gene upregulated with refeeding that was surprisingly not investigated, even though this should be a strong maker of proper meal-induced neuronal activation in the DMV. This is a missed opportunity either to verify the data set or to highlight important limitations if that had been attempted without success.

This is a highly salient point made by the reviewer. Including Fos expression serves as an internal validation of our refeeding condition and the absence of Fos mRNA levels from the original manuscript was an oversight on our part. As shown in our volcano plot, between ad libitum fed and refed mice, there are two significantly *Fos*-associated genes upregulated in the refed group. Therefore, we are confident that the snRNA-Seq analysis accurately captured rapid changes in response to refeeding in the DVC. Only genes differentially expressed (log2 Fold-change >0.5 per group) were considered in the analysis. NS = non-significant.

(5) The focus on transmitter classification is highlighted, but surprisingly, the well-accepted distinction of GABAergic neurons by Slc32a1 was not used, instead, Gad1 and Gad2 were used as GABAergic markers. While this may be proper for the DMV, given numerous findings that Gad1/2 are not proper markers for GABAergic neurons and often co-expressed in glutamatergic populations, this confound should have been addressed to make a case if and why they would be proper markers in the DMV.

The reviewer raises an important point. Indeed, there are discrepancies in expression between the Gad1/2 genes and Slc32a1 gene in other data sets. To analyze this within our data set, we examined the mainly GABAergic magnaclass 1 (see *Slc32a1* UMAP plot below). In magnaclass 1, only 5% and 3% of all neurons exclusively express solely *Slc32a1* without either *Gad1* or *Gad2*, respectively. In line with the reviewer’s comment, we found that 54% of neurons express either *Gad1* or *Gad2* but had no detectable *Slc32a1*. While our failure to detect more cells that co-express *Slc32a1* and Gad genes may be partially due to the low expression of *Slc32a1*, it is also very likely that the DVC, like other brain regions, contains neurons that express the Gad enzymes without co-expression of *Slc32a1*.

This was very much the case with the GLP1 cell cluster, which we identified as the population which had the highest co-expression of excitatory and inhibitory markers. When we refined this analysis to look at expression of excitatory markers with *Slc32a1* (and not other inhibitory genes), there was a marked reduction in the proportion of GLP1 neurons meeting this criterion. We find this is mainly due to the GLP1 cells expressing Gad2 (see plots below). We still find that there are some GLP1-expressing neurons that express excitatory markers and *Slc32a1* and that the GLP1 neurons have a higher proportion of these co-expressing cells than other cell types.

We have extended our results section to reflect this and thank the reviewer for recommending this analysis.

**Author response image 2. sa3fig2:** Slc32a1 expression across all neurons.

**Author response image 3. sa3fig3:** Proportion of neurons in all cell identities expressing glutamatergic markers alone (dark green), Slc32a1 alone (light green), both glutamatergic markers and Slc32a1 (purple) or expressing neither Slc32a1 or glutamatergic markers (grey).

**Author response image 4. sa3fig4:** Balloon plot of Slc32a1, Gad1 and Gad2 across cell types. The GLP1-expressing neurons express Gad2 but minimal Slc32a1.

(6) The Pdgfra IHC as verification is great, but images are not very convincing in distinguishing the 2 (mouse) or 3 (rat) classes of cells. Why not compare Pdgfra and HuC/D co-localization by IHC and snRNAseq data (using the genes for HuC/D) in the mouse and in the rat? That would also clarify how specific HuC/D is for DMV neurons, or if it may also be expressed in non-neuronal populations.

In agreement with the suggestion by the reviewer, we reanalyzed the snRNA-Seq data to identify the extent of the co-expression of HuC/HuD (i.e. *Elavl3* and *Elavl4* genes, respectively) in *Pdgfra*-expressing neurons. The gene expression of the 34 rat neurons belonging to this group are shown in the following heatmap in which each column represents one neuron. As shown, most neurons co-express *Pdgfra* and either HuC or HuD gene. In addition, we shown the UMAP plots of the rat neurons showing expression of the same genes regardless of the neuronal identity assigned. The Pdgfra neurons are visible in darker blue in the last UMAP plot. It's important to note that HuD is a more specific neuronal marker as shown in the table with the average expression of *Elavl3/4* genes, since HuC is expressed by glial cells, specially OPCs and oligodendrocytes. As the HUC/D antibody detects both proteins, this complicates the interpretation of the immunofluorescent staining. While, the snRNA-Seq data suggests these Pdgfra expressing cells are indeed neurons (albeit a rare population), we aim to confirm this in separate studies.

**Author response image 5. sa3fig5:** 

**Author response image 6. sa3fig6:** Average expression (log-normalized counts) of HuC/D by layer 1 cell identity in the rat cells.

**Author response table 1. sa3table1:** 

	connective	glial	neuron	unspecific	vascular
Elavil3 (HuC)	0	2.662964	2.276006	0	0.504195
Elavi4 (HuD)	0.4995	0.375098	5.120204	0.342021	0.723189

(7) The importance of sub-clustering for clusters 23, 26, and 27 is not immediately clear. Does this have any relevance to the mouse vs. rat data? Or fed, fast, refeeding data sets? Or is it just to show the depth that can be achieved?

We appreciate that our justification was not clear within the manuscript. We have clarified our rationale below but briefly, in each case distinct transcriptional profiles were observed, and we pursued this by performing sub-clustering.

Cluster 23 was subclustered as it was found to contain both pre-myelinating and a subset of myelinating oligodendrocytes, therefore, to label them effectively in R instead of cell by cell, those subclusters showing pre-myelinating oligodendrocyte markers were instructed to be labeled as such in the dataset. The remaining cells were labeled as mature oligodendrocytes.

A similar approach was taken for cluster 27 which contained pericytes, endothelial and smooth muscle cells (Figure S5).

In the case of cluster 26, it was possible to find two subclusters of fibroblasts when mapping markers, so they were sub-clustered to instruct in R to label a group with one identity and the other, with the other identity. Therefore, the sub-clustering was done as an aid to label the different identities found through markers mapping (Table S5) in the first clustering round.

All labels were transferred from mouse to rat data using treeArches, including those resulting from the sub-clustering of these clusters. Because this was done to establish identity, it should not be relevant for treatment analyses (e.g. fasted, refed) since they are built from markers that don't change by conditions but remain as identity markers. Indeed, our dataset has an even distribution of these subclusters among samples.